# Imaging the facet surface strain state of supported multi-faceted Pt nanoparticles during reaction

Maxime Dupraz [1,2 ✉], Ni Li[1,2], Jérôme Carnis [2,3], Longfei Wu[2,3], Stéphane Labat[3], Corentin Chatelier[1,2], Rim van de Poll[4], Jan P. Hofmann [4,5], Ehud Almog[6], Steven J. Leake [2], Yves Watier[2], Sergey Lazarev[7], Fabian Westermeier [7], Michael Sprung[7], Emiel J. M. Hensen [4], Olivier Thomas [3], Eugen Rabkin [6] & Marie-Ingrid Richard [1,2 ✉]

Nanostructures with specific crystallographic planes display distinctive physico-chemical properties because of their unique atomic arrangements, resulting in widespread applications in catalysis, energy conversion or sensing. Understanding strain dynamics and their relationship with crystallographic facets have been largely unexplored. Here, we reveal in situ, in three-dimensions and at the nanoscale, the volume, surface and interface strain evolution of single supported platinum nanocrystals during reaction using coherent x-ray diffractive imaging. Interestingly, identical {$hkl$} facets show equivalent catalytic response during non-stoichiometric cycles. Periodic strain variations are rationalised in terms of $O_2$ adsorption or desorption during $O_2$ exposure or CO oxidation under reducing conditions, respectively. During stoichiometric CO oxidation, the strain evolution is, however, no longer facet dependent. Large strain variations are observed in localised areas, in particular in the vicinity of the substrate/particle interface, suggesting a significant influence of the substrate on the reactivity. These findings will improve the understanding of dynamic properties in catalysis and related fields.

[1] Univ. Grenoble Alpes, CEA Grenoble, IRIG, MEM, NRS, 17 rue des Martyrs, 38000 Grenoble, France. [2] ESRF - The European Synchrotron, 71 Avenue des Martyrs, Grenoble 38000, France. [3] Aix Marseille Université, CNRS, Université de Toulon, IM2NP UMR 7334, 13397 Marseille, France. [4] Laboratory for Inorganic Materials and Catalysis, Department of Chemical Engineering and Chemistry, Eindhoven University of Technology, P. O. Box 513, 5600 MB, Eindhoven, The Netherlands. [5] Surface Science Laboratory, Department of Materials and Earth Sciences, Technical University of Darmstadt, Otto-Berndt-Strasse 3, 64287 Darmstadt, Germany. [6] Department of Materials Science and Engineering, Technion-Israel Institute of Technology, 3200003 Haifa, Israel. [7] Deutsches Elektronen-Synchrotron (DESY), 22607 Hamburg, Germany. ✉email: maxime.dupraz@esrf.fr; mrichard@esrf.fr

The chemical properties and performance of supported metallic catalysts depend on their structural properties. The demonstration of the structure-activity relationship on single crystal surfaces[1,2] has motivated the synthesis of shape controlled nanoparticle (NP) systems[3]. Catalytic reactivity and selectivity are strongly influenced by the NP shape and are therefore facet dependent[4,5]. Shaping NPs with well defined exposed facets is an efficient strategy to enhance not only the efficiency of catalysts, but also many other physical properties such as light absorption[6] or electrical conductivity[7]. A better understanding of these facet dependent properties is therefore required to design nanomaterials with enhanced properties. Transmission electron microscopy provides two-dimensional (2D) projected images of NPs with a resolution and sensitivity down to the atomic level[8–10]. When used in situ, the technique can even reveal time-resolved observations of shape or size changes of metal NPs during catalytic processes[11,12]. However, these observations are always challenging to make under *operando* conditions, in particular at elevated temperatures[13] and realistic (1 atm and higher) pressures. In addition to shape-dependent catalytic activity, lattice strain[14] has a considerable influence on the reactivity of metal surfaces. By modifying the distances between surface atoms, tensile or compressive lattice strain induces an upward or downward shift of the d-band centre[15], resulting, respectively, in a strengthening or weakening of the bonding to adsorbed reacting

species. In platinum (Pt) for instance, a 1% tensile (compressive) strain can upshift (downshift) the 5d-band by as much as ~ 0.1 eV[16], dramatically affecting the surface reactivity[17] and in turn the catalytic activity. Moreover, recent results indicate that metal NPs are super-strong and can sustain elastic strains in the range of several percent[18]. This opens a wide window for modifying the catalytic activity of the particles. Correlating surface strain with catalytic performance is therefore of fundamental importance for the development of highly efficient catalytic nanomaterials. Progress in nanomaterial design enables many opportunities for engineering the surface strain in a number of nanocatalysts by tuning structural properties such as composition, architecture (core-shell structure[19,20]) or morphology[21]. Applying a controlled external force on the nanocatalyst to achieve the desired strain state is also a powerful way to get a better insight in the fundamental relationship between surface strain, adsorption properties and reactivity. Several approaches have been proposed to precisely tune surface strain, they include, for instance, the physical bending or stretching of the supporting substrate[22] or extracting and intercalating ions from a battery electrode support[14]. The latter approach has confirmed the clear relationship between enhancement or suppression of Pt oxygen reduction reaction activity under compression or tension, respectively.

If the statement "tension strengthens binding" has gradually become a widely accepted trend, it is not a universal rule[23]. This is particularly true for NPs, where atoms located within facets or at the edges or corners have very different coordination numbers, leading to a large and complex distribution of surface strains[24]. Moreover, the localised nature of adsorption leads to complex site- and adsorbate-dependent responses, emphasising the need to directly study the reaction under strain, rather than extrapolating the known behaviour of individual adsorbates[25]. Nonetheless, studies aiming at investigating the 3D strain evolution in NPs during reaction remain relatively sparse. In particular, the strain evolution at the NP substrate interface and as a function of the nanocrystal facets is mostly unknown. In situ Bragg coherent diffraction imaging (BCDI) recently demonstrated the possibility to acquire 3D strain maps of NPs and nanostructures[26]. In particular, the technique has been successfully used to reveal in situ bulk and surface strain dynamics as well as localisation of active sites during reaction[27–29].

In this work, we demonstrate how the 3D lattice displacement and strain evolution depend on the crystallographic {*hkl*} facets of NPs during reaction. The nanoscale facet dependent strain information presented here has been inaccessible to date.

## Results

**Synthesis and gas reaction**. To reveal structural dynamics, we use a model system consisting of shaped controlled multi-faceted (111)-oriented Pt nanocrystals supported on α-Al$_2$O$_3$ (see Methods section). The sample was placed in a water-cooled gas flow reactor, in which the temperature and gas composition can be tuned (Fig. 1a). The pressure inside the flow reactor was slightly below the atmospheric pressure and a gas flow of 50 ml/min was used. The particle size ranges from 100 nm to 1 μm. Scanning electron microscope images of the sample as well as a histogram of the size distribution of the particles (diameter: (550 ± 230) nm) are displayed in Supplementary Fig. S1. To evaluate size effects, measurements have been carried out on several particles: two 300-nm and 650-nm diameter defect-free nanocrystals, hereafter referred to as NP300 and NP650, respectively, and a third 650 nm NP containing a dislocation (NPD1, Supplementary Fig. S2 and Supplementary Table S3). Each gas cycle typically consisted of four distinct stages, all performed at 450 °C: (1) pure Ar flow, (2) Ar and O$_2$ with O$_2$ concentration

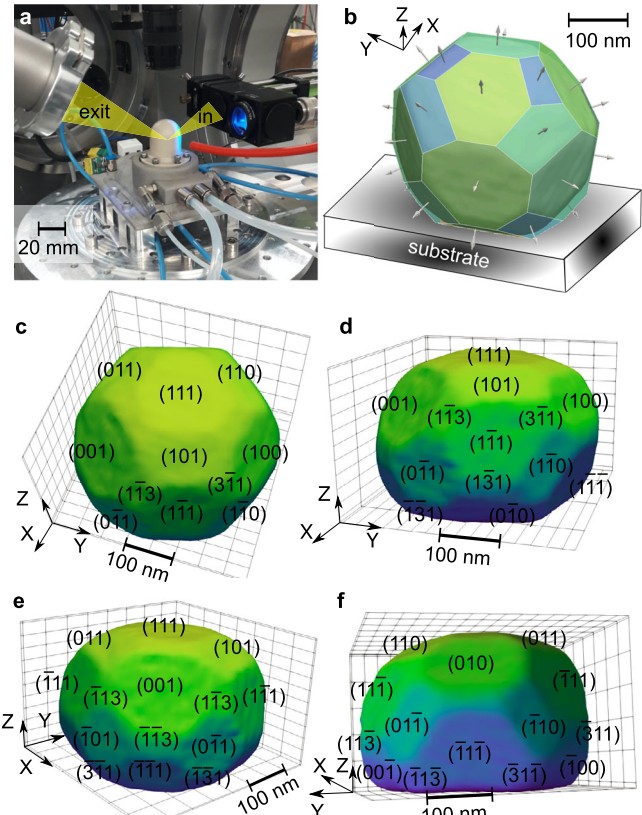

**Fig. 1 Experimental set-up and morphology of Pt nanocrystal.**
**a** Photograph of the experimental set-up with the gas reactor at the P10 beamline of the PETRA III synchrotron. **b–f** Different views of the BCDI reconstruction (drawn at 50% of the maximum Bragg electron density) of NP300. The different facets are indexed by their Miller (*hkl*) indices. The colour coding in Fig. **b** varies as a function of the facet size. The colour coding in Fig. **c–f** varies with the value of the normal of the facets. The position of the substrate with respect to the NP is shown in **b**, showing the [1 1 1] out-of-plane orientation of the Pt NPs.

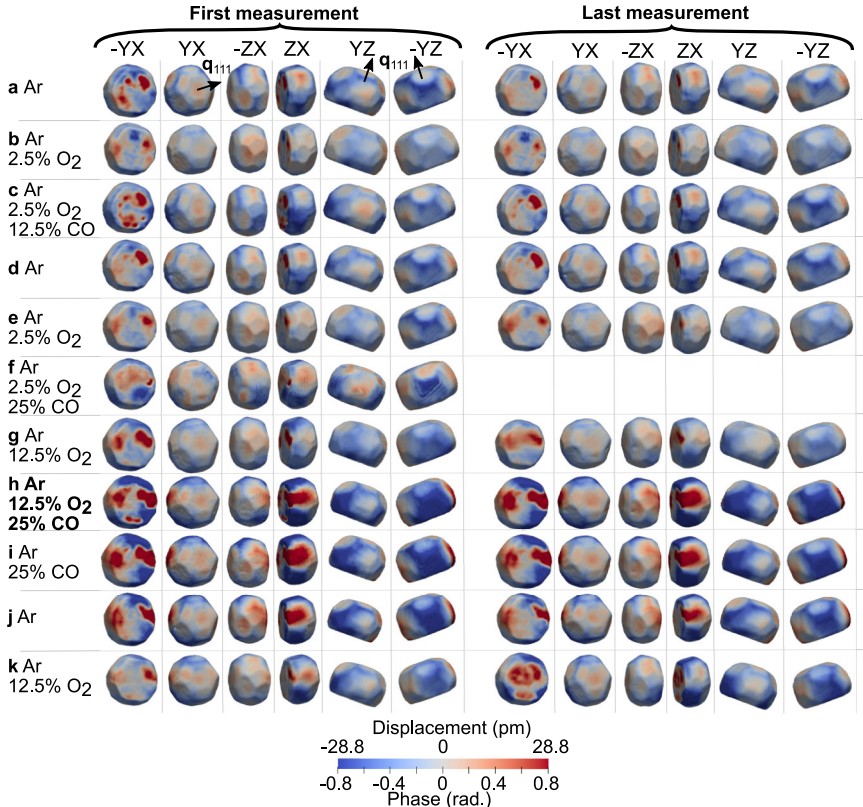

**Fig. 2 In situ 3D lattice displacement images during reaction for NP300.** Evolution of BCDI reconstructions of the displacement field and phase along the [111] direction, drawn at 50% (see Supplementary Fig. S6 for the choice of this value for the isosurface) of the maximum Bragg electron density of NP300 at 450 °C and for different gas mixtures (CO oxidation reaction under stoichiometric conditions is indicated in bold). The left (right) panel, which corresponds to the first (last) measurement, shows the particle at the beginning (end) of the corresponding gas mixture: **a** Ar, **b** Ar + 2.5% O₂, **c** Ar, 2.5% O₂ + 12.5% CO, **d** Ar, **e** Ar + 2.5% O₂, **f** Ar + 2.5% O₂ + 25% CO, **g** Ar + 12.5% O₂, **h** Ar + 12.5% O₂ + 25% CO, **i** Ar + 25% CO, **j** Ar and **k** Ar + 12.5% O₂. The "last" measurement has been taken after reaching the saturation of the changes. It corresponds to the steady state of the particle. Note that the crystal is represented in the laboratory frame, which explains the observed 20° tilt corresponding to the incidence angle of the incoming x-ray beam. In addition, because of the representation in the laboratory frame, X, Y, and Z do not correspond to any particular crystallographic direction. The direction of the scattering vector $q_{111}$ from several points of view is indicated for some images of the top row (condition **a**). During condition **f** only a single measurement was taken and is thus displayed.

ranging from 2% to 12.5%, (3) CO oxidation, either in reducing CO oxidation (2.5% O₂, 25% CO, $\chi = c_{CO} / c_{O_2} = 10$) or in stoichiometric condition (12.5% O₂, 25% CO, $\chi = 2$) and (4) Ar and 25% CO flow. An overview of the gas cycles carried out for each NP is available in Supplementary Tables S1 and S2. A heating ramp of 25 °C/min in pure Ar flow was used to reach the reaction temperature of 450 °C. However, since several scans were measured during the heating ramp, it took 415 min to reach 450 °C from RT (see Supplementary Fig. S3).

**Bragg coherent diffraction imaging.** 3D coherent diffraction patterns were collected at the **111** Pt Bragg reflection using an area pixel detector at an energy of 8 keV at the P10 beamline of the PETRA III synchrotron (see Methods section). Supplementary Fig. S4 displays coherent x-ray diffraction measurements of NP300 for different gas compositions. All the diffraction patterns display well defined streaks indicating the faceted nature of the NP surface. The patterns are all asymmetric, reflecting strain distributions in NP300. Interestingly, large changes are observed in the vicinity of the **111** Bragg peak during stoichiometric CO oxidation (see the first fringes in Supplementary Fig. S4c). These changes arise from local structural variations and suggest that the structure of the Pt particle strongly evolves during CO oxidation.

**Phase retrieval and faceting.** The reconstructed shape of the Pt NP300 particle is displayed in Fig. 1b–f. Phase retrieval was carried out on the diffraction data using the PyNX package[30] (see Methods section). The spatial resolution of the reconstructed images has been evaluated to 10.5 nm using the Phase Retrieval Transfer Function (PRTF)[31,32] (see Supplementary Fig. S5). Well defined facets are observed on the reconstructions. We observed four types of facets for the two nanocrystals: {100}, {110}, {111} and {113}. However, since we only measured the **111** Pt reflection, the angle $\alpha$ between the facet normal **n** and the scattering vector $q_{111}$ needs to be considered to get an accurate picture of the strain dynamics. The {111}, {110} and {113} families are therefore separated in different subfamilies classified from their orientation angle with respect to the scattering vector. Table S4 summarises the different facet subfamilies identified in the two nanocrystals.

**Displacement and strain.** To reveal the local internal structure of the particles during reaction, each diffraction pattern was reconstructed independently. Figures 2, S7, S8 and S9 display the retrieved out-of-plane component of the displacement $u_{111}$ and strain $\epsilon_{111}$ fields for different gas compositions and for the two defect-free NPs. The sensitivity of the BCDI technique to lattice

displacement and strain is discussed in Supplementary Section S12. This allows to follow in 3D and at the nanoscale the structural evolution of the NPs during reaction. Although the measurement of the 3D displacement field and derivation of the full strain tensor requires to probe at least 3 non-coplanar reflections[33], it was decided to measure only a single Bragg reflection due to time constraints. The measurements of several scans for each gas condition and for several particles are indeed extremely time consuming (Supplementary Table S2), making challenging the measurement of multiple Bragg reflections during an in situ experiment. However, we will see in the following that the measurement of a single Bragg reflection is sufficient to extract valuable information on the strain evolution in the presence of gas adsorbates or during reaction, even for the {1$\bar{1}$0} and {11$\bar{3}$} facets, whose normals are respectively perpendicular and almost perpendicular to the **q** vector.

**Average structure evolution.** To underline the average structural changes occurring during reaction for NP300, the full width at half maximum (FWHM) of the **111** Pt diffraction patterns along the $Q_y$ direction, as well as the average or macroscopic strain of the Pt particle, $\langle \epsilon_{111} \rangle$, are shown in Fig. 3 (see Supplementary Fig. S11 for NP650). They have been extracted from the 3D x-ray reciprocal space maps (Supplementary Fig. S4 and Methods) and from the reconstructed strain field (Supplementary Fig. S8). From the nanoscale/local variation of the retrieved strain field, $\epsilon_{111}$ (Supplementary Fig. S8) and displacement, $u_{111}$ (Fig. 2), it was possible to calculate the evolution of the elastic strain energy of the particle as a function of gas flows (Fig. 3c): $E_s = 3K/2 \int (\frac{\partial u_{111}}{\partial x_{111}})^2 dV$ (assumptions of cubic symmetry and isotropic shear-free conditions in the unit cell), where $K$ is the bulk modulus of Pt and $x_{111}$ is the interplanar distance along the [111] direction.

## Discussion

At the beginning of each cycle, under Ar flow, facets and corners/edges of the particles exhibit an opposite displacement (Figs. 2a and S7a), indicating that the facets are in tension, whereas the edges are in compression. To compare this experimental strain with the one expected from a relaxed NP, we have performed molecular statics (MS) simulations taking into account the morphology of the two defect-free NPs (see Methods and Fig. 4). Figure 4 shows a comparison between the experimental $\epsilon_{111}$ strain field measured in NP650 in Ar atmosphere and the $\epsilon_{111}$ strain field obtained by energy minimisation of a 200 nm NP with a similar shape. The Foiles Pt potential[34], which yields the best agreement with the experimental data, was used to model the interaction between Pt atoms. In order to enable a quantitative comparison with the measured experimental surface strain field, the same voxel size (6.7 nm) was used for the experimental and simulated NPs. The latter corresponds to an average of the voxel size along the three directions. This implies that the surface strain is actually averaged over the 30 topmost surface layers. The surface strain also depends, to some extent, on the selection of the isosurface on the experimental data, as discussed in Supplementary Fig. S6 and in ref. [35]. Free surface boundary conditions were used for all facets, except at the interface ($\bar{1}\bar{1}1$) plane, where a −0.05% compressive strain was imposed in order to mimic the thermoelastic strain induced by the substrate. A remarkable quantitative agreement is obtained with the experimental data. The tensile strain measured experimentally on the {11$\bar{3}$} and {100} facets is well reproduced by the simulation, while the compressive strain measured on the {1$\bar{1}$1} facets is also well captured, as also confirmed by Supplementary Table S5, which compares the average strain per facet between the experimental

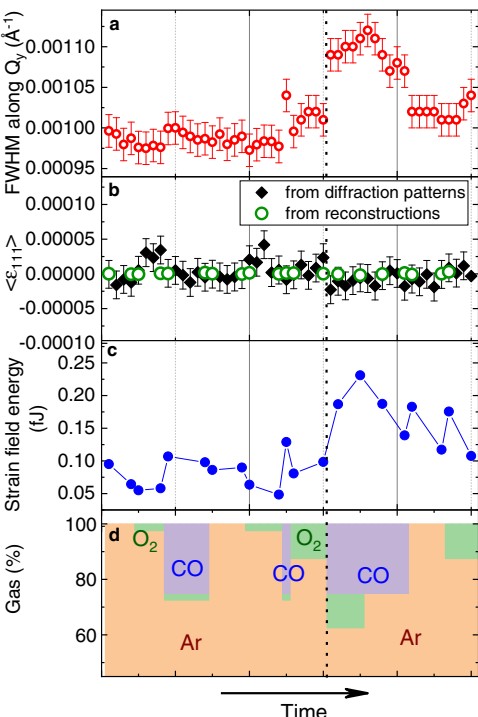

**Fig. 3 In situ structural evolution of NP300 during reaction.** Evolution of (**a**) the full width at half maximum (FWHM) of its diffraction pattern along the $Q_y$ direction, of (**b**) its average strain, $\langle \epsilon_{111} \rangle$ calculated from the centre of mass of the diffraction patterns or from the reconstructed strain field, of (**c**) its strain field energy and of (**d**) the gas composition at the inlet (Ar, $O_2$, and CO are in orange, green and purple, respectively) as a function of time. The dashed line indicates the beginning of the CO oxidation reaction under stoichiometric conditions.

Ar state and MS calculations. Even more convincing, the precise location of the tensile and compressive regions is also accurately reproduced by the simulation. Based on this simulation, we can conclude that the measured experimental strain field depends not only on the element, but also on the NP shape and boundary conditions. The reducing conditions before Ar exposure (Supplementary Table S1) rule out the presence of $O_2$ adsorbates on the free surface. CO adsorbates on the other hand, in particular, on the edges and corners cannot be completely excluded even at this high temperature[36]. However, the remarkable agreement with the strain distribution obtained by MS calculations (see Fig. 4 for NP650 and Supplementary Fig. S12 for NP300) suggests that the NPs are mostly free of adsorbates at the beginning of each cycle in Ar, as shown by ref. [37].

An order of magnitude estimate of the strain can be made assuming that the surface stress $\sigma$ is counterbalanced by the internal stress in the particle: $<\epsilon> = 4<\sigma>/(dE)$ (1), where angular brackets refer to average values, $d$ is the particle diameter, and $E$ is the isotropic Young modulus of Pt. Estimating $<\sigma>$ by the surface energy of Pt yields $<\epsilon> \sim 3 \times 10^{-4}$ and $1 \times 10^{-4}$ for the large simulated and experimentally studied particles (NP650), respectively, in a good agreement with the strain amplitudes in Fig. 4. Highly inhomogeneous strain distribution over different facets in simulated particle can be attributed to the difference in surface stresses of different facets, to the elastic anisotropy of Pt, and to the elastic singularities at the facet edges and corners. Indeed, the surface stresses acting on a sharp edge where the two facets meet produce elastic singularity similar to the singularity of the elastic stress filed of a lattice dislocation. Such singularities promote higher strains for the facets of smaller lateral size (note

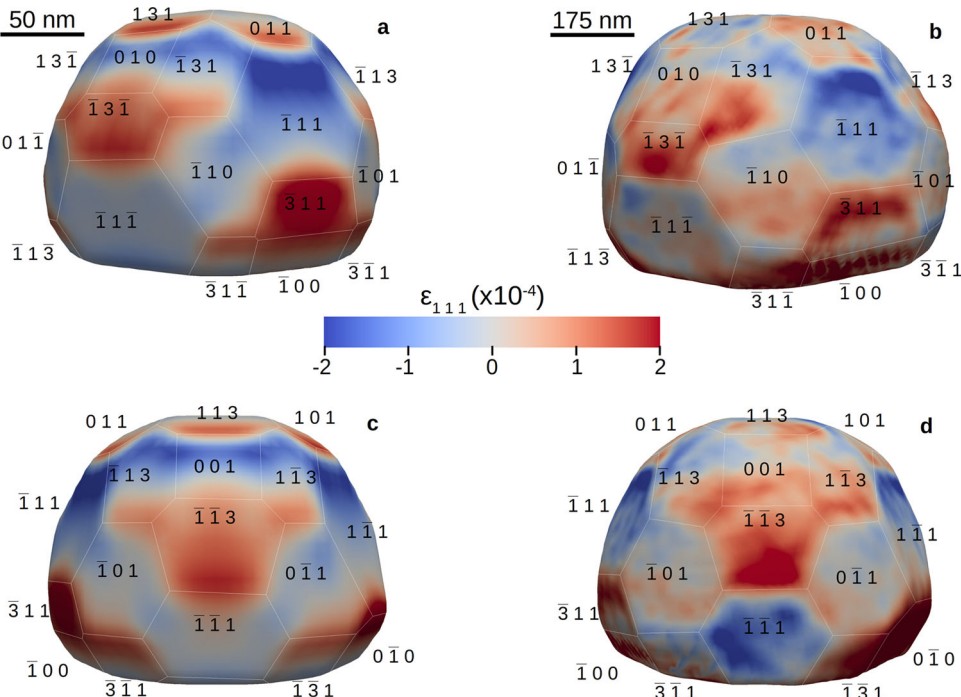

**Fig. 4 Comparison between experiment and simulation for NP650. a, c** 200 nm simulated NP with the same faceting as NP650 and relaxed by energy minimisation seen from two different field of views (**b**), (**d**) Experimental $\epsilon_{111}$ strain field measured in Ar atmosphere for NP650 and seen from the same field of views.

that the small {113} facets in Fig. 4 exhibit higher average strain than their larger counterparts). In the experimentally studied particles, the mismatch stress associated with the particle-substrate interface is an additional factor contributing to the stress inhomogeneity. The above simple estimate (1) demonstrates that the strain evolution during $O_2$ adsorption and CO oxidation observed in the present work (see Fig. 2) is consistent with adsorption- and oxidation-induced changes of the surface stress and, possibly, with the reconstruction of the facet edges and corners changing the modalities of stress singularities.

During reaction, the in situ monitoring of the average strain $\langle \epsilon_{111} \rangle$ (Fig. 3b) shows little change of the macroscopic strain for NP300, while its FWHM along $Q_y$ and its elastic strain energy (Fig. 3a, c) clearly increase during CO oxidation. This indicates local structural evolutions during reaction, well illustrated in Figs. 2 and S8, where structural heterogeneities are observed at the surface of the NP.

One of the strengths of BCDI resides in its ability to probe the structural evolution at the catalyst/support interface. Such information cannot be obtained by in situ TEM techniques. Interestingly, our results show that not only the surface of the particle is impacted by the chemical reactions but also the particle/support interface. Indeed, large variations of the displacement or strain field are observed at the interface between the catalyst and the support (substrate) for both defect-free NPs. The cooling down to room-temperature (RT) after thermal dewetting at 1100 °C induces a large thermoelastic strain at the NP/substrate interface. The latter can be partially relaxed by the formation of an interfacial dislocation network, which evolves dynamically during heating of the sample, giving rise to compressive regions at the interface. During reaction, the observed large lattice displacement and strain changes could result from gas diffusion at the interface through the dislocation network, which would act as diffusion channels for the adsorbates[38].

The reaction induced strain evolution is not only observed at the interface and free surfaces but also in the volume of the NPs.

As illustrated by the 2D slices of the reconstructed displacement and strain of NP300 shown in Supplementary Figs. S13 and S14, the displacement and strain fields induced by the binding of adsorbates propagate in the particle interior, as demonstrated in refs. [12,28,29]. Interestingly, both strain and displacement fields are extremely stable and reproducible for a given gas condition for NP650 (Supplementary Figs. S7 and S9). Since for each gas condition, the first scan was typically measured 10–15 min after changing the gas mixture, this implies that the stabilisation time/ kinetics of the reaction are less than 10 min for this NP. In contrast, the displacement field tends to evolve dynamically between two measurements, mostly at the interface, in the same gas atmosphere for NP300 (Fig. 2). During the stoichiometric reaction, for instance, it required more than 90 min before reaching a stable surface strain state (Fig. 2h). This difference in behaviour between the two defect-free NPs could be attributed to two factors: their difference in size, NP650 being much larger than NP300, or the different faceting/surface morphology. As shown in Supplementary Table S4, NP650 is indeed much more faceted than NP300, with a larger number of high-index {113} facets (20 in NP650 vs 9 in NP300), making the former closer to the thermodynamical equilibrium shape of Pt[39–41].

The strain variation induced by the binding of adsorbates on the free surfaces depends on the facet type[42]. Using the facet analyser plugin[43] of ParaView (see Methods), one can visualise the average surface displacement and strain per facet in 3D (Supplementary Fig. S15). Figure 5 shows the details of the facet dependent strain evolution during the first two reducing CO oxidation cycles (Fig. 5a–h) and during stoichiometric CO oxidation (Fig. 5i–l) for NP300. The strain variations per facet with respect to the previous state ($\Delta\langle\epsilon_{111}\rangle$) are also indicated. During the first two cycles with a 2.5% $O_2$ flow, the strain evolution is strongly dependent on the *hkl* Miller indices of the facets. Interestingly, all facet families except the {1$\bar{1}$0} facets (where a slight tensile strain builds up) experience a strain relaxation during $O_2$ exposure, either a relaxation of the compressive strain

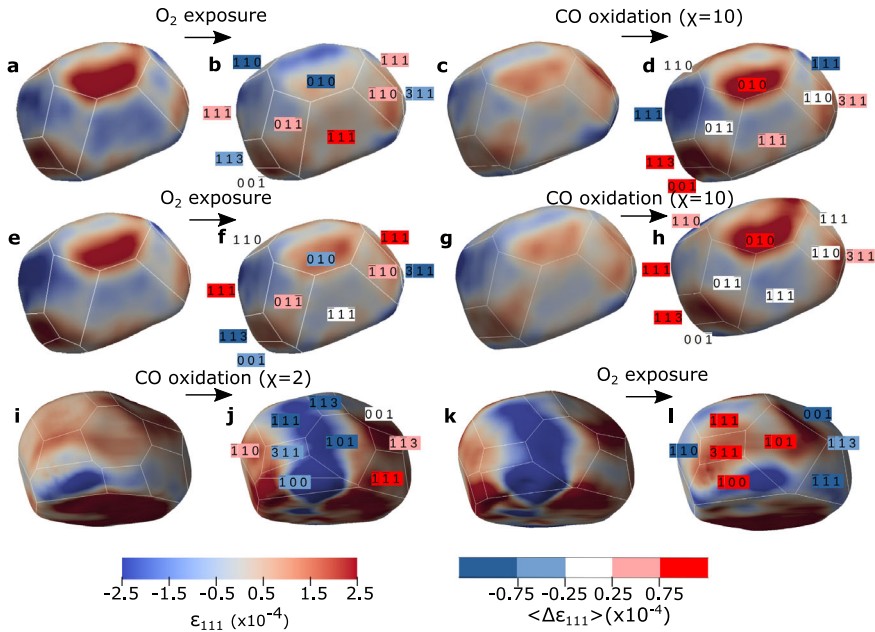

**Fig. 5 Strain evolution during O$_2$ adsorption and CO oxidation for NP300. a–d** 1st cycle: **a** Ar, **b** Ar + O$_2$: 2.5% (14 min. exposure), **c** Ar + O$_2$: 2.5% (59 min. exposure) and **d** Ar + CO: 25% + O$_2$: 2.5% (10 min. exposure). **e–h** 2nd cycle: **e** Ar, **f** Ar + O$_2$: 2.5% (10 min. exposure), **g** Ar + O$_2$: 2.5% (76 min. exposure) and **h** Ar + CO: 25% + O$_2$: 2.5% (19 min. exposure). **i–l** 3rd cycle: **i** Ar + O$_2$: 12.5% (70 min. exposure), **j** Ar + CO: 25% + O$_2$: 12.5% (21 min. exposure), **k** Ar + CO: 25% + O$_2$: 12.5% (53 min. exposure) and **l** Ar + O$_2$: 12.5% (13 min. exposure). The colour coding of the Miller indices denotes the variation of the average strain per facet $\Delta\langle\epsilon_{111}\rangle$ with respect to the previous state.

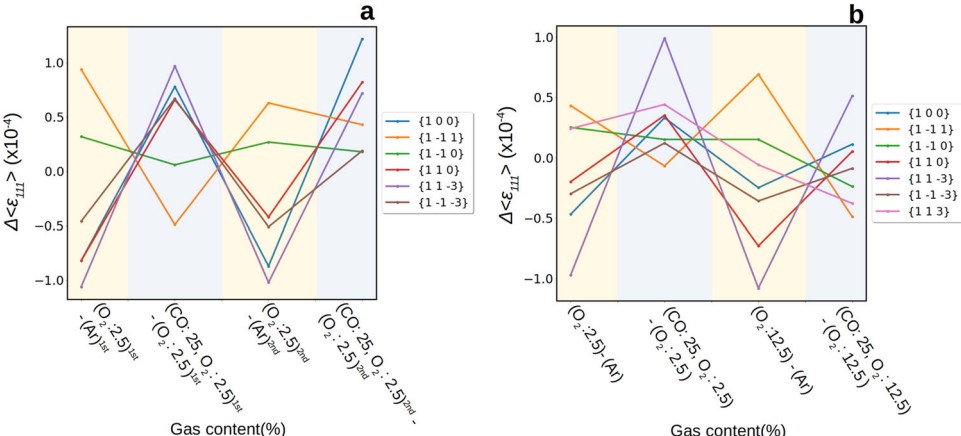

**Fig. 6 Comparison of the average strain evolution ($\Delta\epsilon_{111}$) per {$hkl$} facet subfamily during O$_2$ adsorption and CO oxidation.** The strain variation is given between two consecutive gas conditions. **a** NP300, first two cycles ($\chi = 10$). **b** NP650 ($\chi = 10$) and ($\chi = 2$). A yellow background indicates a switch from pure Ar to Ar + O$_2$ while a blue background indicates a switch from Ar + O$_2$ to CO oxidation reaction conditions.

for the {1$\bar{1}$1}-type facets or a relaxation of the surface tensile strain for the {100}, {11$\bar{3}$}, {1$\bar{1}$3} and {110} facets (see also Supplementary Fig. S16 and Supplementary Table S6). The opposite trend is observed during CO oxidation in CO excess ($\chi = 10$), where most of the facets come back to their initial strain state. Figure 6a, which represents the strain variation between two consecutive gas conditions, illustrates this cyclic behaviour, which is also observed for NP650 (Fig. 6b). Albeit the presence of a surface oxide can not be completely ruled out, density functional theory (DFT) calculations have shown that surface oxide, including $\alpha$-PtO$_2$ are not stable under the reaction conditions of this work[37]. Rather, the facet dependent reactivity can be rationalised in terms of oxygen chemisorption during O$_2$ exposure and oxygen reduction/desorption during reducing CO oxidation

condition. Upon oxygen exposure, oxygen adsorption will occur preferentially on edge and corner atoms followed by the facet sites with increasing coordination number, $n$: {113} and {110} ($n = 7$), {100} ($n = 8$) and {111} ($n = 9$). The initial surface tensile strain in the {11$\bar{3}$}, {1$\bar{1}$3} and {100} facets also favours adsorption, which induces a compressive strain on these facets during O$_2$ exposure at 450 °C. In contrast, the high coordination number of the {111} surfaces and their initial compressive strain make the binding of adsorbates on these facet sites energetically unfavourable. Binding is mostly restricted to edge and corner atoms for the {1$\bar{1}$1} surfaces (also supported by DFT calculations - see Supplementary Table S8), which results in an inward displacement of the corner and edge atoms compensated by an outward displacement of the facet atoms (tensile strain or relaxation of the compressive strain)

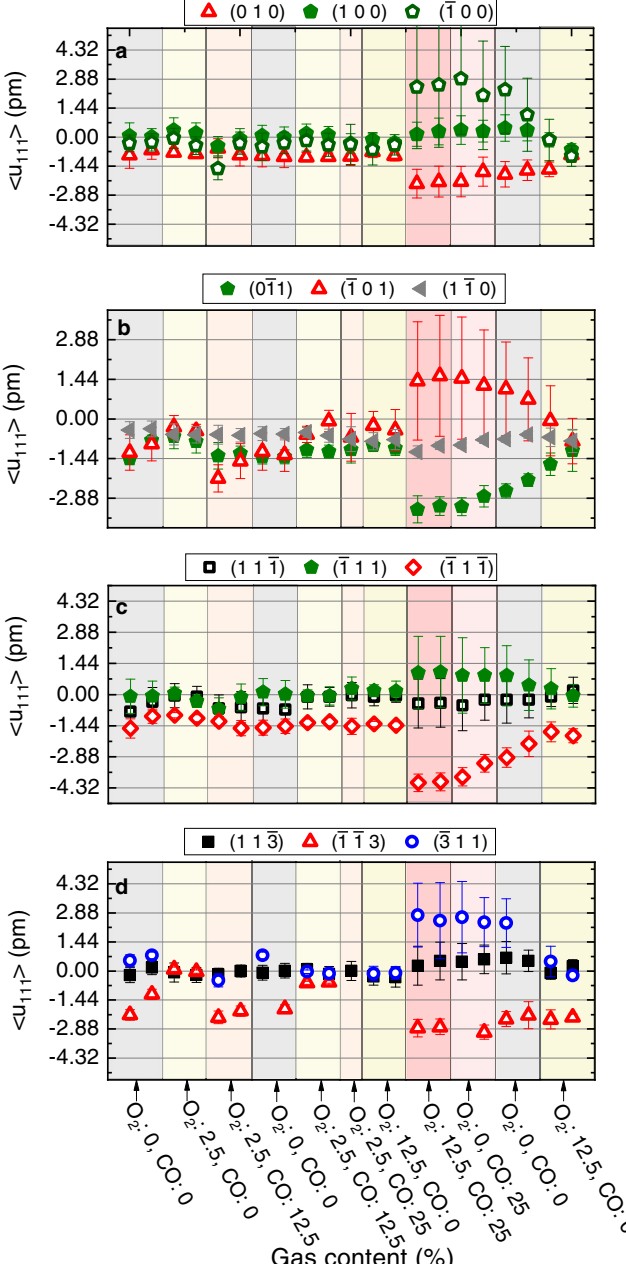

**Fig. 7 Facet analysis of the displacement field for NP300.** Evolution of the average displacement field for three selected facets of the four different facet subfamilies: **a** {100}, **b** {1$\bar{1}$0}, **c** {1$\bar{1}$1} and **d** {11$\bar{3}$} facets. The evolution of three facets are shown with at least one of them experiencing large variations during the CO oxidation and one of them unaffected by the reaction. The error bars represent the standard deviation of the displacement per facet. The large inhomogeneous displacement observed during the stoichiometric CO oxidation reaction is reflected by large error bars.

to lessen the surface area[44]. During reducing CO oxidation ($\chi = 10$), the almost perfect reversibility of the strain state indicates a complete oxygen reduction, leaving only metallic Pt on the surfaces.

Figures 6 and S17 and Table S7 reveal that the facet dependent character of the strain evolution is mostly retained during stoichiometric CO oxidation in NP650. However, this is not the case for NP300 (see Supplementary Fig. S18 and Table S9), where the facet dependence is no longer observed during stoichiometric CO

oxidation. Figure 7 displays the evolution of the average lattice displacement per facet in NP300. For the sake of clarity, only 3 facets are shown for each family. The trends for each individual facet are shown in Fig. S21 (displacement) and S22 (strain). Interestingly, one can observe that at least one facet per family experiences very large variations, while one of them is mostly unaffected during stoichiometric CO oxidation. In particular, a localised region, hereafter referred to as the ZX region (see also Fig. S19), including at least one facet belonging to the different subfamilies (($\bar{1}$00), ($\bar{3}$11), ($\bar{1}$01), ($\bar{1}$11) and ($1\bar{1}3$)) exhibits a very large positive displacement (Figs. 2h and 7) corresponding to a compressive strain (Fig. 5j, k). This region is surrounded by two regions (labelled YZ and -YZ), where a large negative displacement, corresponding mostly to tensile strain, builds up (Figs. 2h and S8h). Together with large variations of their average surface displacement, the facets located in these regions also exhibit a large increase of the standard deviation of the displacement (Fig. 7), demonstrating the inhomogeneous character of the reaction induced surface displacement. On the other hand, the region on the other side of the nanocrystal, including the (11$\bar{3}$), (11$\bar{1}$), and (10$\bar{1}$) facets, for instance, appears less affected by the reaction. At odds with previous works on single crystal surfaces, the analysis of the average strain evolution confirms that the latter is weakly correlated to the facet type during stoichiometric CO oxidation. Facets belonging to the same family can exhibit a different reactivity during stoichiometric CO oxidation (see Supplementary Fig. S18), in good agreement with the recent observations of ref. [29]. Even more interesting, the facet position with respect to the interface also affects its reactivity. This is illustrated in Supplementary Fig. S23 and Table S10, where the average displacement evolves dramatically for the facets in contact with the substrate, while the changes in the facets close to the top surface are barely noticeable. This observation is confirmed in Supplementary Fig. S19, where the largest strain variations are observed in the (00$\bar{1}$) and (11$\bar{3}$) facets in contact with the substrate. The measurements carried out in stoichiometric conditions suggest that the substrate/particle interaction has a major impact on $CO_2$ formation. Although mass spectrometry data was not collected for this sample, Supplementary Fig. S24 displays results from a different sample containing Pt NPs with a very similar size distribution confirms the formation of $CO_2$ during CO oxidation at 450 °C, further suggesting that the structural changes observed during reaction are indeed linked to $CO_2$ formation. These findings strengthen the importance of taking into account metal-support interaction to improve performance in catalysis in agreement with ref. [45]. It is highly probable that $CO_2$ formation is increased at these preferred active sites. Strain may lead to the occurrence of fluctuating surface steps acting as active sites at high temperature (450 °C). Reaction induced structural changes are associated to a higher reactivity and could improve the activity of the nanocatalyst.

Another question lies in the origin of the very large strain variations observed during stoichiometric CO oxidation in NP300 but not in NP650. Not only the magnitude but also the inhomogeneity of surface and bulk strains increase. These reaction induced structural changes are characteristic of high activity regime, which suggests a higher catalytic activity in NP300[12,27,28]. A closer look at Supplementary Table S9 also reveals that if the tensile strain increases in some regions of NP300, all families of facet experience on average a compressive strain during stoichiometric CO oxidation. Ref. [12] suggested that diffusion of oxygen into the subsurface and bulk can lead to the formation of vacancies, which create disorder, leading to an increase of both bulk and surface strain. This diffusion of oxygen adsorbates in the lattice could be at the origin of the increase of the FWHM of the Bragg peak (Fig. 3a) in Ar and 12.5% $O_2$. The strain induced

reactant diffusion can be relieved by surface reconstruction or roughening, leading to the multiplication of step sites, where CO adsorption is favourable in these experimental conditions[11]. These CO adsorbates would be at the origin of the large inhomogeneous and predominantly compressive strain observed during stoichiometric CO oxidation (Figs. 2h and 5j, k). Switching back to $O_2$ induces a relaxation of most of the large surface strain that built-up during the stoichiometric oxidation (Supplementary Figs. S18, S19 and S25), as confirmed by the decrease of the FWHM of the displacement histogram (i.e., the microstrain, Fig. 3a) and of the strain field energy (Fig. 3c). This large strain relaxation would result from the oxidation of the CO adsorbates on the step sites[11]. The localised character of the strain variations could also be ascribed to the presence of localised oxide islands for instance on the {100} facets[46]. A lowering of the CO oxidation barriers at the boundaries between the {100} facets surface oxide islands and the metallic {111} facets can lead to enhanced activity and would explain the local strain variations[47]. This relaxation is also accompanied by a decrease of the extent of the compressive regions at the NP/substrate interface (Fig. 2i, j and Supplementary Fig. S8i, j) suggesting that a reorganisation of the interfacial dislocation network can also contribute to accommodate the large strain induced during reaction.

In summary, we have studied in situ the 3D lattice displacement and strain dynamics in Pt NPs under different gas compositions at a temperature of 450 °C using BCDI. Different *hkl* surface morphologies have been evaluated under identical reaction conditions. During reaction, only local modifications of the strain and displacement fields are observed, while the average/macroscopic lattice strain does not evolve. The strong local structural changes are associated to chemical interactions, which are facet and/or support-dependent based upon the type of reaction. Under reducing CO oxidation conditions, the reactivity is mostly independent of the NP size and exhibits a strong dependence on the facet type. During stoichiometric CO oxidation, on the other hand, the strain evolution depends on the nanocrystal size and faceting. While it is still cyclic and facet dependent in NP650, this is no longer the case in NP300, where large strain variations are observed in localised areas, most of them being in the vicinity of the NP/substrate interface. This work provides clues for the active sites during reaction for supported Pt NPs and could contribute to a better understanding of the relationship between facet-related surface strain and chemistry during reaction. In addition to the applications-oriented interest, our in situ observation of how an individual nanoparticle "breathes" during CO oxidation reaction is unprecedented in its picometre-scale displacement resolution, and will motivate deep theoretical works on the fundamental, quantum-level mechanisms of catalysis.

## Methods

**Growth**. Pt nanocrystals were prepared by the solid-state dewetting of a 30-nm thin Pt film for 24 h at 1100 °C in air. The Pt film was deposited on $\alpha$-$Al_2O_3$ (sapphire) with an electron beam evaporator. The Pt nanocrystals have their *c*-axis oriented along the [111] direction normal to the (0001) sapphire substrate. A standard photolithography method was employed to prepare a patterned layer of photoresist on sapphire prior to the electron beam evaporation of Pt. The lithographic processing route ensured that a number of dewetted Pt particles are well-separated from their neighbours and that only one crystallite is irradiated by the incoming x-ray beam.

**Average lattice strain calculation**. The 3D datasets of the reciprocal space maps (RSMs) of the NP give enough information to retrieve the average strain of the measured (111) atomic planes as a function of the gas composition (Ar, $O_2$ or CO). The average strain is related to the variation of the *d*-spacing of the (111) Pt atomic planes: $\langle \epsilon_{111} \rangle = (d_{111}^{meas.} - d_{111}^{ref.})/d_{111}^{ref.}$, where $d_{111}^{ref.}$ is the reference $d_{111}$ spacing (here, first measured value at 450 °C under Ar) and $d_{111}^{meas.}$ is calculated from the measured $\mathbf{q}_{111}$ scattering vector: $d_{111}^{meas.} = 2\pi/\| \mathbf{q}_{111}\| = 2\pi/\sqrt{q_x^2 + q_y^2 + q_z^2}$, where $q_x$, $q_y$ and $q_z$

are the coordinates of the $\mathbf{q}_{111}$ scattering vector. The conversion of pixels from the detector images into $\mathbf{q}_{111}$ coordinates was done using the xrayutilities package[48]. The $\mathbf{q}_{111}$ coordinates were determined by taking the centre of mass of the 3D datasets, while the FWHM of the **111** Pt diffraction patterns along the $Q_y$ direction was obtained by performing a Gaussian fit of the 3D data sets.

**Bragg coherent diffraction imaging**. The BCDI experiment was performed at a beam energy of 8 keV (wavelength of 1.55 Å) at the P10 beamline of the PETRA III synchrotron at DESY; X, Y and Z being along the incident beam (downstream), outboard and vertically upward, respectively. The beam size was focused down to 0.9 μm (vertically) × 1.1 μm (horizontally) using Be compound refractive lenses. The sample was mounted in a dedicated water-cooled gas-flow chamber with the substrate surface oriented in the horizontal plane on a high-precision (1 μm) XYZ-stage that was mounted on a 6-circle Huber diffractometer. The diffracted beam was recorded with a 2D EIGER X4M photon-counting detector (2070 × 2167 pixels with a pixel size of 75 × 75 μm) positioned on the detector arm at a distance of 1.839 m. We measured the **111** Pt Bragg reflection in three-dimensions by rotating the particle around the Bragg angle (20°) through 3°, with 0.012° steps for NP300 and through 2°, with 0.005° steps for NP650. The counting time per point was 0.5 s. The detector was positioned at an out-of-plane angle of 39.26°.

**Phase retrieval algorithm**. The reconstructed Bragg electron density and phase were obtained using the PyNX package[49]. Phase retrieval was carried out on the raw diffracted intensity data. The initial support, which is the constraint in direct space, was estimated from the auto-correlation of the diffraction intensity. A series of 1400 Relaxed Averaged Alternating Reflections (RAAR[50]) plus 300 Error-Reduction (ER[51,52]) steps, including shrink wrap algorithm every 20 iterations[53], were used. A support was built from the best reconstructed object. Afterwards, the support was fixed during the first 20 iterations and then let free, while reapplying a series of 600 Hybrid Input Output (HIO[54]) followed by 900 Relaxed Averaged Alternating Reflections plus 300 Error-Reduction steps, the shrink wrap algorithm was applied every 20 iterations. The phasing process included a partial coherence algorithm to account for the partially incoherent incoming wave front[55]. To ensure the best reconstruction possible, we kept only the best 50 reconstructions (with lowest free Log-Likelihood[56]) from 300 with random phase start and performed the decomposition into modes[56]. The reconstruction was then corrected for refraction using the bcdi package[57]. After removing the phase ramp and phase offset (calculated at the centre of mass of the support), the data was finally interpolated onto an orthogonal grid for ease of visualisation.

**Molecular statics**. In order to model accurately the experimental NPs, several material simulation tools were used. The simulated NPs were generated using the atomistic simulation code MERLIN (Rodney, D. Merlin in a nutshell. Unpublished (2010)), by creating a cube of atoms and cutting it along the <1 1 1>, <1 0 0>, <1 1 0> and <1 1 3> crystallographic directions, the position of the cut planes being defined by the ratio of the surface energies $\gamma_{111}/\gamma_{100}$, $\gamma_{110}/\gamma_{100}$ and $\gamma_{113}/\gamma_{100}$. Note that different ratios were used for the two NPs, both of them differing from the theoretical surface energies for Pt[39], implying that the nanocrystals did not reach their thermodynamical equilibrium shape. Several crystal sizes were considered ranging from 80 × 80 × 80 to 500 × 500 × 500 unit cells. This corresponds to a size of 32 nm/937000 atoms for the smallest configurations and 200 nm/230 x $10^6$ atoms for the largest configurations. To obtain accurate and realistic relaxed configurations that reproduce as faithfully as possible the displacement fields measured in the experimental particle, MS simulations were carried out with the open-source Large-scale Atomic/Molecular Massively Parallel Simulator (LAMMPS)[58]. The interaction between atoms were modelled with different embedded-atom model (EAM) Pt potentials which predict slightly different elastic properties and surface energies, parameters that are essential to model accurately the experimental displacement and strain fields. We used the EAM potentials developed by refs. [34,59,60]. The NPs were relaxed at 0 Kelvin using a conjugate gradient algorithm to obtain the equilibrium displacement field.

In order to allow a quantitative comparison with the experimental data, the three-dimensional diffraction patterns were calculated by summing the amplitudes scattered by each atom with its phase factor, following a kinematic approximation:

$$I(\mathbf{q}) = |\textstyle\sum_j f_j(\mathbf{q})e^{-2\pi i\mathbf{q}\cdot\mathbf{r}_j}|^2, \qquad (1)$$

where $\mathbf{q}$ is the scattering vector, $f_j(\mathbf{q})$ and $r_j$ are respectively the atomic scattering factor and position of atom $j$. The computation was performed with a GPU using the PyNX[61] scattering package, which considerably sped up the calculation of the 3D diffraction patterns. The reciprocal volume over which the calculation was performed was selected based on the target voxel size in the real space data. For the largest configurations (200 nm, 230 × $10^6$ atoms), the voxel size was equal to the experimental one (6.7 nm). The surface strain values for both NPs shown in Supplementary Table S5 are obtained for a NP size of 200 nm and a voxel size equal to the experimental one.

Finally, the real space displacement and strain fields, $u_{111}$ and $\epsilon_{111}$ were derived from the complex sample density $\tilde{\rho}(\mathbf{r})$. The latter was obtained by performing a

simple inverse Fourier transform of the scattered amplitude $\tilde{A}(\mathbf{q})$:

$$\tilde{\rho}(\mathbf{r}) = \rho(\mathbf{r})e^{2\pi i\mathbf{g}.\mathbf{u}(\mathbf{r})} = FT^{-1}[\tilde{A}(\mathbf{q})], \qquad (2)$$

The projection of the displacement field $\mathbf{g}_{111}.\mathbf{u}_{111}$ and the corresponding strain field $\epsilon_{111} = \frac{\partial u_{111}}{\partial x_{111}}$ could then be directly compared to the measured experimental strain field as shown in Supplementary Fig. S25.

**Facet analysis.** We have used the Facet Analyser plugin of the ParaView software, which is an automated 3D facet recognition algorithm that allows to detect the voxels at the surface of the particles and to index their facets by tuning parameters like sample size, angle uncertainty, splat radius or minimum relative facet size[43]. The algorithm is based on the analysis of the probability distributions of the orientations of triangle normals of mesh representations of the particles. From the output list (voxels and associated global orientation) of the facet recognition algorithm, we have developed a Python script to determine the *hkl* Miller indices of each facet as well as their average lattice displacement, strain and associated standard deviation.

**DFT calculations.** DFT calculations were performed using the Vienna Ab Initio Simulation package (VASP), within the projector-augmented wave method (PAW), for the description of the interaction between the valence electrons and the ionic core. The GGA-PBE functional was used, with DFT-D3 corrections. We consider atomic valences to be 2s2 2p4 (O) and 5d9 6s1 (Pt). Total energies were minimised until the energy differences were less than $1 \times 10^{-6}$ eV between two electronic cycles. Atomic structures were relaxed until the Hellmann–Feynman forces were as low as 0.02 eV/Å. Calculations were performed using a 450 eV cutoff energy and Γ-centred $14 \times 14 \times 14$ and $14 \times 14 \times 1$ Monkhorst-Pack grids for bulk and (100)/(111) surface calculations, respectively. Those parameters were chosen to achieve a precision for the total energy lower than 0.1 meV/atom. In this work, the approach used to compute the adsorption of oxygen onto Pt surfaces is the typical symmetric slab mode, wherein a supercell of the crystal oriented to expose its (*hkl*) surface is generated, and atoms are removed from a portion of the supercell to create a vacuum (thick symmetric slabs of 20–30 Å, void thickness ~ 20 Å). The atomic strain tensor for each atom in the slab was extracted using the atomic strain modifier[62] in OVITO, considering a cut-off radius of 3.09 Å. This value corresponds to the first minimum of the pair distribution function for Pt, i.e., halfway between the first and the second shell of neighbours. To compare with the experimental values, the atomic strain values were averaged over the size of the slab along the z-direction (20 Å). Since the average surface strain mostly scales with the voxel size, this value was then multiplied by the ratio between the size of the slab and the experimental voxel size (67 Å). Finally, to allow a quantitative comparison with the experimental values $\langle \epsilon_{111} \rangle$ of the $\{1\bar{1}1\}$ and $\{100\}$ facets, the theoretical $\langle \epsilon_{111} \rangle$ values have to be multiplied by $\cos^2\alpha$ where $\alpha$ is the angle between $\mathbf{q}_{111}$ and the facet normal (see Supplementary Table S4). This assumption is reasonable if the atoms relax perpendicular to the facets, which is the case in the DFT simulations. However, it is also obvious from Fig. 4 that the strain field strongly depends on the NP shape and boundary conditions, which results in large atomic displacements not only perpendicular to the facets but also parallel to the facets (for instance for the $\{11\bar{3}\}$ facets, that can not be captured by the DFT calculations).

## Data availability
The data supporting the findings of this work are available from the corresponding author on reasonable request. When published the data will be deposited in the CXI database (https://www.cxidb.org/).

## Code availability
The phasing algorithm PyNX[30] is available at http://ftp.esrf.fr/pub/scisoft/PyNX/. The scripts used for BCDI data post-processing, PRTF calculations and stereographic projections belong to the bcdi package[57] (https://doi.org/10.5281/zenodo.3257616), that can be downloaded from PyPI (https://pypi.org/project/bcdi/) or GitHub (https://github.com/carnisj/bcdi).

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

## Acknowledgements

The authors are grateful to PETRA synchrotron for allocating beamtime. The measurement was performed at the P10 beamline of the PETRA III synchrotron at DESY, a member of the Helmholtz Association (HGF). We thank P10 beamline staff for excellent support during the experiment. M.-I.R. acknowledges funding from the European Research Council (ERC) under the European Union's Horizon 2020 research and innovation programme (grant agreement No. 818823). The research leading to this result has been supported by the project CALIPSOplus under the Grant Agreement 730872 from the EU Framework Programme for Research and Innovation HORIZON 2020. M.-I.R and E.R. also wish to thank the support by a grant from the Ministry of Science and Technology, Israel and from the Centre National de la Recherche Scientifique (CNRS), France.

## Author contributions

M.D., N.L., J.C., L.W., S. Lab, R.v.d.P., S. Lea, S. Laz, F.W., M.S. and M.-I.R. carried out the experiment. M.-I.R. directed the project. M.D., N.L. and M.-I.R. analysed the data. M.D. performed the MS simulations. C.C. carried out DFT simulations. E.A. and E.R. prepared the samples. Y.W. designed and built the gas reactor. M.D. and M.-I.R. wrote the manuscript. J.P.H., E.H. and O.T. helped in designing the project. All authors reviewed, discussed the manuscript and have given approval to its final version.

## Competing interests

The authors declare no competing interests.
