## [Peer Review File · Nature Communications]

Title: Imaging the facet surface strain state of supported multi-faceted Pt nanoparticles during reactionREVIEWER COMMENTS

Reviewer #1 (Remarks to the Author):

This paper reports Bragg coherent X-ray diffraction imaging of multi-faceted Pt nanoparticles during the non-stoichiometric and stoichiometric CO oxidation cycles. They observed facet-dependent and periodic strain variations during the non-stoichiometric cycles. However, no significant facet dependence was observed during the stoichiometric cycles. They observed that the displacement and strain fields of the 650 nm-sized nanoparticle show no significant changes with different gas conditions.

The present study provides valuable information on Pt nanoparticles as a catalyst for the shape and structural changes in situ, i.e., during the catalytic reactions, not available from any other techniques. Therefore I think that it is worth to be published.

However, I have the following points that need to be clarified.

1. They showed the gas cycles in the Table S1. It is clearly their logbook.

It is better to show how much time spend in each cycle. However, it is still not clear why they ran those cycles at a particular time. They seemed not to keep the same duration for the comparison with the different sizes. In addition, do not the changes they observed show the time dependence?

2. It is not clear the size dependence they claimed. It seemed that NP300 and NP650 were on the same substrate, meaning that the NP650 was measured late but exposed already. Please explain the experimental process with initial changes. The authors mentioned that the NP650 was compared with the 200 nm simulated nanoparticle relaxed by energy minimization. Is this related to reactivity?

3. Since they measured Pt 111 Bragg peak, in principle, they should not observe any component of 111 projection to 1-10.

4. Overall, it is somewhat misleading from their presentations of the displacement and strain depending on the facet, i.e., Figs. 2, S3, S4, S5, etc. Even though they mentioned the displacement and strain are the projection of the 111 direction and XY, etc. was indicated on top of the figure, XYZ coordinate is not clearly indicated with the crystal orientation as well as the measured orientation. It probably needs an arrow to indicate measured Q in the figure with relevant coordinates. It is better to show “before” and “after” in the figure.

5. They did show the displacements and strain at the surface of the images. Please explain how they observed and the meaning.

6. In Figure 2a, the displacements are changes in (-YX) even in the Ar environment. Is it due to the thermal relation of the initial strain? But it is again changed in Fig.2b (before), which means that even 2.5% of O₂ affects the strain? If that is the case, it is a somewhat different observation that the oxygen adsorbs the edges than the surfaces.

Reviewer #2 (Remarks to the Author):

In this work, the authors present an in situ three-dimensional study of strain evolution of platinum

nanocrystals under various gas atmospheres with coherent x-ray diffractive imaging. The 3D strain distribution in catalytic nanoparticles is important for the understanding of the catalytic activity. They claim identical facets show equivalent catalytic response during non-stoichiometric cycles of CO oxidation. During stoichiometric conditions, the strain evolution is no longer facet dependent and large strain variations are observed in the vicinity of the substrate/particle interface. This manuscript provides important insights however, the manuscript needs further and necessary clarifications prior to publication. The following initial questions and comments should be addressed.

- While the present study is structurally relevant and is a potentially interesting example of advanced and powerful tool for materials characterisation, I have some reservation from the catalytic point of view. A catalyst must be active, possibly at low temperature and stable. Much smaller platinum nanoparticles, when present in supported catalyst in the form of few nm size are more active (and at lower temperature), while larger Pt nanoparticles present lower activity and CO conversion occurs at medium high temperature. Moreover, although gas mixtures relevant for the CO oxidation reaction are used in this work, there is absolutely no clue of the reaction occurrence with such big nanoparticles. Therefore, I ask the authors to modify the title considering that 300 and 650 nm platinum nanoparticles are far from relevant catalysts for the catalytic reaction.
- Use of supports to keep separated and well dispersed Pt nanoparticles is mandatory for obtaining high and stable activity. Moreover, metal support interface has been shown to be extremely important for the activity. At the beginning of the manuscript, first figure, indicate the position of the oxide support on the 3D reconstructed objects.
- NP300 and NP650 seem to be directly synthesised on the same substrate. What is the dispersion in size of the Pt particles on the Al₂O₃ support? Are NP300 and NP650 representative of the sample? It might be good to show electron microscopy images of the supported sample with the well-separated Pt particles as explained in the methods section.
- What are the gas flows, heating ramps used in this work? Without such information, the experiment cannot be reproduced. Moreover, it can directly alter the catalytic performances.
- How long does one rocking curve take? Considering the times mentioned into the manuscript/figures, we could estimate this duration, but it would be good to make it clearer for the reader. Are the same parameters used for both sizes of nanoparticles?
- For most of the gas mixtures, two BCDI measurements have been performed and compared (beginning/end). Is the “end” condition, a “stable” condition? Did the author check the time before stabilisation (or even deactivation)? Please comment on this.
- Why nanoparticles 1 and 2 are presented in the table S1, while they are neither shown nor discussed in the manuscript. What are the effects of the previous cycles on the studied NP (NP300 and NP650)? Is there any ageing effect on the catalytic material? Please comment on possible effects on the strain of

19h hours separating the two cycles investigated by BCDI for NP650.

- Although obtained in different experiments, it would be interesting to show the evolution of catalytic performances of the sample during different cycles. Indeed, the type/number of cycles might directly alter the strain distributions observed within the BCDI study.
- Are the catalytic performances affected in the same way between the stoichiometric and non-stoichiometric conditions for both sizes of NP? Can it be correlated to the differences observed on the displacement field and strain distributions?
- Did the authors investigate “fresh” samples (without previous cycles)? What did they observe?
- Regarding the MS simulations, the authors simulated NP of ~ 32 and 200 nm. Please provide details about the 32 nm simulations. Also why choosing a 200 nm NP for the simulation? Please comment on the difference of sizes between the simulations and experiments. Does the size affect the average strain obtained on the simulated NP? Please add a new figure for NP300 with the comparison between experiment and simulation (like figure 4 but for NP300).
- Did the author observe the formation of an interfacial dislocation network, as mentioned page 7? Please comment and identify those dislocations.
- Overall, there are too many references. Their number should be reduced, with a better selection. Please check typing for the following references: 11; 13; 26; 29; 31; 35; 38; 50; 51.
- Although the whole figures cannot be presented in the main manuscript, I would suggest showing only the ones concerning NP300 in the main manuscript and the ones of NP650 in supplementary. Mixing both NP in the figures of the manuscript/supplementary is too confusing.
- Figure 3,b, please modify the symbols used for $\langle \epsilon_{111} \rangle$ from diffraction patterns and from reconstruction, we can hardly distinguish them.
- Figure 4. What is shown in a/c and b/d? the same simulated NP and reconstructed NP with different field of view? Or beginning / end of the gas mixture? Please modify the figure caption.
- What are the corresponding conditions of Figure S8?
- In the conclusion: “our operando observation of how an individual nanoparticle “breathes” during CO oxidation reaction is unprecedented in its picometer-scale displacement resolution”

This work is far from being an operando study. As described in Nature Reviews Materials, 3 (2018) 324: “operando means studying the catalyst under the true reaction conditions, while in situ refers to both model and true reaction conditions. An operando study also requires the simultaneous online analysis of

catalyst performance (activity and/or selectivity) using, for example, mass spectrometry (MS) or gas chromatography (GC) to measure product concentrations.”

Thus, this study can be considered as an in situ study, but due to the lack of online analysis of catalyst performance, this is not an operando study.

Please comment and modify the conclusion.

Reviewer #3 (Remarks to the Author):

The manuscript reports on the study of strain in faceted Pt nanocrystals under controlled reaction conditions using Bragg coherent diffraction imaging, which has been developed over the years into a quantitative method for imaging 3D strain at the spatial resolution tens of nm as the cited references show (Refs. 38-44). This work concerns the development of strain and its facet dependence. Two nanocrystals of size 350 and 600 nm were investigated over multiple diffraction scans and a combination of gas reaction conditions.

The major findings are related to the observation of strain and displacement fields and dependence on reaction conditions. The discussions on size-dependence are less convincing the sampling regarding particle size is very limited and the size of the measured particles generally are of hundreds of nm, which are large for nanoparticles. An extension of the study to smaller nanoparticles could be helpful, but it is not clear what the smallest nanoparticles that could be investigated by this technique.

The strain and displacement fields are presented in figs. 2, 4 and 5 as three-dimensional surface color plots. Are these the measured surface strain and displacements? What does “drawn at 50% of ...” mean in fig. 2 caption?

The Pt nanoparticles were formed by annealing in air at 1100 C. Why in air? Does Pt oxidize under such condition? Is heating at 450 C under argon sufficient to drive out oxygen for a comparison with MD calculation (the agreement between theory and exp is remarkably good, could this indicate an insensitivity of the measured strain to actual surface structure?).

The surface referred here presumably has a thickness, rather than single or few atomic layers that are studied by microscopy techniques. The measured strain is for the 111 lattice plane and thus does not provide information the full strain fields. What is measured and presented should be made clearer to the readers.

The selection of (111) reflection rather than (200) or (113) is likely coming out the convenience of the experiment. Whether such condition is suitable for the study of surface strain in a nanoparticle with different facets should be discussed.

REVIEWER COMMENTS

Reviewer #1 (Remarks to the Author):

This paper reports Bragg coherent X-ray diffraction imaging of multi-faceted Pt nanoparticles during the non-stoichiometric and stoichiometric CO oxidation cycles. They observed facet-dependent and periodic strain variations during the non-stoichiometric cycles. However, no significant facet dependence was observed during the stoichiometric cycles. They observed that the displacement and strain fields of the 650 nm-sized nanoparticle show no significant changes with different gas conditions. The present study provides valuable information on Pt nanoparticles as a catalyst for the shape and structural changes in situ, i.e., during the catalytic reactions, not available from any other techniques. Therefore I think that it is worth to be published.

Answer: We thank the reviewer for this positive evaluation of the manuscript.

However, I have the following points that need to be clarified.

1. They showed the gas cycles in the Table S1. It is clearly their logbook. It is better to show how much time spend in each cycle.

Answer: We modified the table to show the time spent in each cycle rather than the cumulative time. For the sake of clarity and to get a more accurate picture of the history of each nanoparticle (NP), we also numbered the stoichiometric cycles (SC), reducing cycles (RC) and oxidizing cycles (OC).

However, it is still not clear why they ran those cycles at a particular time. They seemed not to keep the same duration for the comparison with the different sizes. In addition, do not the changes they observed show the time dependence?

Answer: We tried to be consistent in our methodology to collect the data. For instance, the first scan for a given gas condition was typically measured around 10-15 minutes after we switched gas conditions. The idea behind this measurement strategy is related to the time resolution of the technique. For a given NP, the exposure time and angular range were selected to optimize the spatial resolution (high signal to noise ratio over a large q range, q being the scattering vector) and to measure a sufficiently large volume of the reciprocal space along the scanning direction. Moreover, the angular steps need to be small enough, in order to fulfill the oversampling conditions [1] along the scanning direction for phase retrieval. Overall, there is a trade-off between enhancing the spatial resolution and reducing the time of measurement.

In order to achieve a similar spatial resolution for the two NPs (NP300 with a diameter of 300 nm and NP650 with a diameter of 650 nm), a longer exposure time was used for NP300 (0.5s for NP650 vs 2s for NP300 in the first part of the experiment, i.e. from scan 890 to scan 998; same exposure time (0.5s) but less attenuators in the incident beam after scan 998). A typical rocking curve took 7 minutes for NP650 and 11 minutes for NP300. As the technique is not able to capture changes on the timescale of the seconds but rather on the timescale of few minutes, we decided to wait 10-15 minutes before the first measurement at a given gas condition such that the NPs can reach a steady state.

For NP650, we observed that the strain measured for a given gas condition does not show any time dependence. For NP300, however, Figures 2, S7 and S11 reveal that some changes occur, even after more than one hour in the same gas atmosphere. For each gas condition, we therefore measured at least three scans and made sure that the strain field was no longer evolving before changing the gas mixture. This has obviously some implications on the time spent for each gas condition and explain why the measurements carried out on NP300 during the first stoichiometric CO oxidation took longer (2h, from scan 1016 to scan S1034) than the measurements for the over gas conditions (typically between 60 and 90 minutes).

2. It is not clear the size dependence they claimed. It seemed that NP300 and NP650 were on the same substrate, meaning that the NP650 was measured late but exposed already. Please explain the experimental process with initial changes.

Answer: All particles were indeed on the same substrate, meaning that both particles were exposed to several stoichiometric and reducing cycles before they were measured, as described in Table S1. We also measured a third particle (NP1) during the first stoichiometric cycles. Contrary to NP300 and NP650, this particle contains a dislocation with an extended displacement field, which makes more difficult to follow the evolution of the surface displacement field. However, the strain evolution during oxygen exposure and stoichiometric CO oxidation appears to be consistent with the O₂ adsorption / desorption mechanism described in the manuscript (strain relaxation during oxygen exposure followed by an increase of the surface strain during stoichiometric and non-stoichiometric CO oxidation).
→ We added a figure (Supplementary Figure S9) illustrating the strain relaxation in O₂ atmosphere.

Regarding the influence of the previous stoichiometric cycles on the strain evolution, we cannot completely exclude that NP650 also experienced the large changes observed in NP300 during SC4 and SC5 (from scan 1016 to scan 1034 and from scan 1105 to scan 1130 in the table). However, the surface strain states during conditions (a) and (h) in NP650 are remarkably similar, in contrast with NP300 which never recovers its initial strain state after the first stoichiometric cycle (from scan 1016 to scan 1034). Although several stoichiometric oxidations were carried out between the two conditions, NP650 manages to recover its initial strain state. This suggests that NP650 did not experience the large changes reported in NP300 during the first stoichiometric CO oxidation but also that the surface of NP650 is mostly free of adsorbates during condition (h).

Finally, regarding the size dependence, the present work clearly demonstrates different trends between NP300 and NP650 during stoichiometric CO oxidation. We attributed these differences to a size effect, but it might as well be related to the difference in surface termination (*i.e.* faceting: for example, contrary to NP300, NP650 shows small {131} facets at its top) between the two NPs. This is now discussed in the Results section.

The authors mentioned that the NP650 was compared with the 200 nm simulated nanoparticle relaxed by energy minimization. Is this related to reactivity?

Answer: The comparison of the initial strain state with the MD simulation is relevant because the NP is supposed to be free of adsorbates before the reaction. The NPs are indeed exposed to reducing conditions before each cycle, ensuring that their surface is in principle free of oxygen adsorbates. In addition, we do not expect a significant CO coverage at this high temperature [2]. For this reason, a MS simulation with free surfaces as boundary conditions and a strain imposed to the interface simulating the thermoelastic strain induced by the sapphire substrate should give a reasonable approximation of the strain state experienced by the NP. The excellent agreement between the experimental surface strain state and the surface strain state obtained from the simulation suggests that the NPs are indeed free of adsorbates in their initial state.

With our computing capabilities, it was not possible to simulate a 650 nm NP and the largest simulated NP is 200 nm in size. The strain being averaged over the voxel size, the maxima of the surface strain largely depends on the choice of this quantity. Hence, by imposing the experimental voxel size on the simulated data one can obtain a good quantitative agreement between the simulation and experiment. This excellent agreement also demonstrates the quality of the reconstructed data. However, this simulation is not meant to give insight in the relationship between strain evolution and reactivity. This could be the object of a future work using reactive MD simulations (with larger constraints on the size that can be simulated).

3. Since they measured Pt 111 Bragg peak, in principle, they should not observe any component of 111 projection to 1-10.

Answer: This would be true if the relaxation of atoms would occur only perpendicular to the facet (out-of-plane), which can be a reasonable approximation when dealing with extended flat free surfaces (from a thin film for instance). However, this approximation does not hold for NP even for the sizes considered in this work. The strain distribution is indeed strongly influenced by the shape of the NP. This is well captured by the MS simulations, where large strains are observed on the {1 1 -3} and {1 -1 0} facets (even if the normal of the {1 -1 0} facets is perpendicular to the [1 1 1] direction). This demonstrates that large in-plane displacements (strains), *i.e.* parallel to the facets, are also observed in the vicinity of these facets. These in-plane displacements, parallel to the surface have a large

projection onto the 1 1 1 scattering vector and are therefore predicted in the simulated data and measured in the experimental data.

4. Overall, it is somewhat misleading from their presentations of the displacement and strain depending on the facet, i.e., Figs. 2, S3, S4, S5, etc. Even though they mentioned the displacement and strain are the projection of the 111 direction and XY, etc. was indicated on top of the figure, XYZ coordinate is not clearly indicated with the crystal orientation as well as the measured orientation. It probably needs an arrow to indicate measured Q in the figure with relevant coordinates.

Answer: We added an arrow in these figures to show the direction of the Q vector for some selected viewpoints. Note that the crystal is shown in the laboratory frame and not in the crystal frame, therefore the X, Y and Z directions do not correspond to a particular crystallographic direction.

→ This is now mentioned in the caption of Figs. 2, S6-S8.

It is better to show "before" and "after" in the figure.

Answer: This statement would be inaccurate since both scans were measured after gas exposure (typically 10-15 minutes for the first scan shown in the left panel, see also S1 and 60 to 90 minutes for the last scan shown in the right panel). We slightly modified the caption such that it is now clearly stated that the left(right) panels correspond to the first and last measurement respectively for each gas condition.

5. They did show the displacements and strain at the surface of the images. Please explain how they observed and the meaning.

Answer: BCDI allows to reconstruct a 3D complex image of isolated objects such as NPs. Its amplitude corresponds to the electron density of the object, while its phase corresponds to a projection of the 3D displacement field $\mathbf{u}(\mathbf{r})$ onto the scattering vector ($\mathbf{g} = 1\ 1\ 1$ in the present work).

In the kinematic approximation of scattering, which is fully justified here because we study small crystals ($<1\ \mu\text{m}$), the scattered intensity is the square modulus of the Fourier transform of the atomic scattering factor:

$$I(\mathbf{q}) = \left| \int f(\mathbf{r}) e^{2i\pi\mathbf{q}\cdot\mathbf{r}} \right|^2, \quad (1)$$

where the integration is performed over the illuminated volume. The Bragg geometry probes the crystalline order and coherent X-rays can be used in Bragg geometry to investigate the deviation of the sample from a perfect crystal order.

For an imperfect crystal, one can define \mathbf{r}_0 the positions of a perfect lattice that approximates the crystal and $\mathbf{u}(\mathbf{r})$ the displacement of the atoms from the perfect lattice such that $\mathbf{r} = \mathbf{r}_0 + \mathbf{u}(\mathbf{r})$ (see Fig. 1).

Now let's consider a Bragg reflection with a reciprocal space vector \mathbf{g} (defined on the perfect lattice that approximates the crystal). We focus on a region of the reciprocal space (RS) in the vicinity of \mathbf{g} : the phase factor defined in the exponential of eq. (1) can be decomposed as follows:

$$\mathbf{q}\cdot\mathbf{r} = \mathbf{q}\cdot\mathbf{r}_0 + \mathbf{g}\cdot\mathbf{u}(\mathbf{r}) + (\mathbf{q} - \mathbf{g})\cdot\mathbf{u}(\mathbf{r}), \quad (2)$$

The third term in Eq. (2) can be neglected if $|(\mathbf{q} - \mathbf{g})\cdot\mathbf{u}(\mathbf{r})| \ll 1$ (Takagi's approximation), which is equivalent to assuming small distortions of the lattice and a restricted extent of the RS, two perfectly reasonable assumptions in the present work. This gives:

$$I(\mathbf{q}) = |FT[\tilde{f}(\mathbf{r})]|^2 \text{ with } \tilde{f}(\mathbf{r}) = f(\mathbf{r}) e^{2i\pi\mathbf{g}\cdot\mathbf{u}(\mathbf{r})}, \quad (3)$$

In the present work, we consider only non-resonant scattering, such that the atomic scattering factor $f(\mathbf{r})$ is essentially the electron density of the sample $\rho(\mathbf{r})$, while the modified scattering factor $\tilde{f}(\mathbf{r})$ is referred as the complex electron density $\tilde{\rho}(\mathbf{r})$: its modulus is the physical electron density while its phase encodes the projection of the displacement field onto the diffraction vector \mathbf{g} .

This displacement field $\mathbf{u}(\mathbf{r})$ contained in the phase term: $\varphi = \mathbf{g} \cdot \mathbf{u}(\mathbf{r})$ can be understood by considering a block of material which is displaced from the rest of the lattice by a vector $\mathbf{u}(\mathbf{r})$ as illustrated in Fig. 1. The phase of the X-ray wave scattered by this block of atoms is shifted relative to the rest of the reference crystal by an amount $\varphi = \mathbf{g} \cdot \mathbf{u}(\mathbf{r})$; provided that a complex image of the sample is obtained, the phase shift appears in the reconstructed image as a region of complex density with the same amplitude but a different phase:

Figure 1 Sensitivity of coherent X-rays to lattice displacements (from Robinson & Harder, 2009, ref. [3])

The strain is then simply a derivative of the reconstructed displacement. For more information, please refer to refs [3,4].

→ We added a brief Supplementary Materials note explaining how displacement and strain can be derived from BCDI (Supplementary Materials S6).

6. In Figure 2a, the displacements are changes in (-YX) even in the Ar environment. Is it due to the thermal relation of the initial strain?

Answer: It is correct that some changes, in particular at the interface in (-YX) are observed during condition (a: 100% Ar) in Fig. 2. As seen in Fig. 2a, the displacement field mostly evolves at the NP/substrate interface in Ar atmosphere. These changes in Ar atmosphere are even more pronounced after the stoichiometric cycle (condition (g) to (i)) and are attributed to a reorganization of the interfacial dislocation network to relax the thermoelastic strain.

But it is again changed in Fig.2b (before), which means that even 2.5% of O₂ affects the strain? If that is the case, it is a somewhat different observation that the oxygen adsorbs the edges than the surfaces.

Answer: 2.5% of O₂ affects the strain. The changes in the surface strain are rationalized in terms of oxygen chemisorption during O₂ exposure (and oxygen/desorption during reaction condition). We believe that the adsorption preferentially occurs at the edge and corner atoms followed by the facet sites with increasing coordination number, which would be consistent with previous observations from the literature [5]. This is discussed in pages 8 and 9 of the manuscript:

“Rather, the facet dependent reactivity can be rationalized in terms of oxygen chemisorption during O₂ exposure and oxygen reduction/desorption during reducing CO oxidation condition. Upon oxygen exposure, oxygen adsorption will occur preferentially on edge and corner atoms followed by the facet sites with increasing coordination number [...] Binding is mostly restricted to edge and corner atoms for the {1 -1 1} surfaces (also supported by DFT calculations see Table S8), which results in an inward

displacement of the corner and edge atoms compensated by an outward displacement of the facet atoms (tensile strain or relaxation of the compressive strain) to lessen the surface area.”

The changes at the nanoparticle/substrate interface can be explained by O₂ diffusion at the interface, where a dislocation network may occur.

Reviewer #2 (Remarks to the Author):

In this work, the authors present an in situ three-dimensional study of strain evolution of platinum nanocrystals under various gas atmospheres with coherent x-ray diffractive imaging. The 3D strain distribution in catalytic nanoparticles is important for the understanding of the catalytic activity. They claim identical facets show equivalent catalytic response during non-stoichiometric cycles of CO oxidation. During stoichiometric conditions, the strain evolution is no longer facet dependent and large strain variations are observed in the vicinity of the substrate/particle interface. This manuscript provides important insights however, the manuscript needs further and necessary clarifications prior to publication. The following initial questions and comments should be addressed.

Answer: We thank the reviewer for this positive evaluation of the manuscript.

•1) While the present study is structurally relevant and is a potentially interesting example of advanced and powerful tool for materials characterisation, I have some reservation from the catalytic point of view. A catalyst must be active, possibly at low temperature and stable. Much smaller platinum nanoparticles, when present in supported catalyst in the form of few nm size are more active (and at lower temperature), while larger Pt nanoparticles present lower activity and CO conversion occurs at medium high temperature. Moreover, although gas mixtures relevant for the CO oxidation reaction are used in this work, there is absolutely no clue of the reaction occurrence with such big nanoparticles. Therefore, I ask the authors to modify the title considering that 300 and 650 nm platinum nanoparticles are far from relevant catalysts for the catalytic reaction.

Answer: Although we agree with the referee that the NPs measured in this work are very large, we believe that the results presented in the manuscript are still relevant and can contribute to gain a better understanding of the reaction mechanisms in the much smaller catalysts (<10 nm) typically used for the reaction.

Unfortunately, we were not able to use a mass spectrometer for this experiment in order to measure the reaction products. However, measurements carried out on a very similar sample (solid state dewetted Pt NPs on a sapphire substrate), have clearly evidenced CO₂ production (see Figure Supplementary Materials S22 attached from Ref. [5]).

In addition, we believe that the large changes observed in NP300 during stoichiometric CO oxidation are a clear indication of the occurrence of the reaction since they are clearly not related to thermal/thermoelastic effects and are not observed during measurements carried out in any other gas condition. These two elements strongly suggest the occurrence of the reaction even in these large NPs. For these two reasons, we disagree with the reviewer and would prefer to leave the title of this manuscript as it is.

•2) Use of supports to keep separated and well dispersed Pt nanoparticles is mandatory for obtaining high and stable activity. Moreover, metal support interface has been shown to be extremely important for the activity. At the beginning of the manuscript, first figure, indicate the position of the oxide support on the 3D reconstructed objects.

Answer: As described in the methods section, the lithographic processing route ensured that a number of dewetted Pt particles are well separated from their neighbours. We also modified the first figure such that the position of the oxide support is now indicated.

•3) NP300 and NP650 seem to be directly synthesised on the same substrate. What is the dispersion in size of the Pt particles on the Al₂O₃ support? Are NP300 and NP650 representative of the sample? It might be good to show electron microscopy images of the supported sample with the well-separated Pt particles as explained in the methods section.

Answer: We added a SEM picture of the sample showing the dispersion of the NPs on the substrate (Figure S2). As illustrated in Fig. S2, the NP size ranges between 100 and 800 nm. Note that we also measured a third NP which is also 650 nm in size (NP1). Few diffraction patterns (DPs) and reconstructions of NP1 measured for several gas conditions are now shown in Supplementary Figure S9. The morphology of the three NPs is consistent with the SEM images, they are therefore representative of the sample.

•4) What are the gas flows, heating ramps used in this work? Without such information, the experiment cannot be reproduced. Moreover, it can directly alter the catalytic performances.

Answer: We used a gas flow of 50 ml/min throughout the experiment and used a heating ramp of 25°C/min. However, since several scans were measured during the heating of the sample, it actually took a total of 415 minutes to reach the reaction temperature (450°C from RT). This is now explicitly written in the manuscript. We also added a Supplementary Figure showing the details of the heating ramp (Figure S1).

•5) How long does one rocking curve take? Considering the times mentioned into the manuscript/figures, we could estimate this duration, but it would be good to make it clearer for the reader. Are the same parameters used for both sizes of nanoparticles?

Answer: The duration of the rocking curves indeed depends on the NP considered. For a given NP, the exposure time and angular range were selected to optimize the spatial resolution (high signal to noise ratio over a large q range) and to measure a sufficiently large volume of the reciprocal space along the scanning direction. Moreover, the angular steps need to be sufficiently fine in order to fulfill the oversampling conditions along the scanning direction for phase retrieval. Overall, there is therefore a trade-off between enhancing the spatial resolution and reducing the time required to perform the measurement. Based on these considerations, a typical rocking curve took 7 minutes for NP650 (2° range, 0.005° steps, 0.5s per point) and 11 minutes for NP300 (3° range, 0.012° steps, 2s per point). Note that from S998 (condition **g** for NP300), we decided to remove some of the filters in the incoming beam path and decrease the exposure time to 0.5s per point in order to speed up the measurement process. With these settings, the statistics were roughly the same before (6 filters, 2s exposure: 70000-80000 cts/s) and after (1 filter, 0.5s exposure: 80000-100000 cts/s). This allowed to decrease the time required to measure a rocking curve to 4 minutes 30s.

•6) For most of the gas mixtures, two BCDI measurements have been performed and compared (beginning/end). Is the "end" condition, a "stable" condition? Did the author check the time before stabilisation (or even deactivation)? Please comment on this.

Answer: As seen from Figs. S6 and S8, the 3D displacement and strain fields are extremely reproducible between the first and last scan of each gas conditions for NP650. This suggests that both the first and last measurements correspond to stable conditions for this NP. Since the first scan was typically measured 10-15 minutes after switching gas condition, this implies that the stabilization time is less than 10 minutes.

For NP300 the situation is slightly different as illustrated in Figs.2, S7 and S11. Indeed, there are several conditions where the strain state measured during the first scan of a given gas condition differs from the strain state measured during the last scan of the same gas condition. This is particularly visible for conditions **c**, **e**, **g**, **h** and **k** (Fig. 2). Therefore, we typically repeated 4 to 5 scans per gas condition, ensuring that the last two were reproducible before switching the gas condition. Overall the stabilization is clearly longer for NP300 and takes more than one hour for several gas conditions. It is unclear at this point if this can be attributed to size effects or if it is related to the morphology of the NP. Indeed NP650 is more faceted than NP300 (significantly larger number of {1 1 3}-type facets for instance) and is closer to the Pt thermodynamical equilibrium shape [7,9]. This difference in surface morphology could also contribute to explain the different behavior of the two NPs. This is now discussed in the discussion section of the manuscript:

“This difference in behavior between the two NPs could be attributed to two factors: the difference in size between the two NPs, NP650 being much larger than NP300 or the different faceting / surface morphology. As shown in Supplementary Table S2, NP650 is indeed much more faceted than NP300, with a larger number of high-index {113} facets (20 in NP650 vs 9 in NP300), making the former closer to the thermodynamical equilibrium shape of Pt than the latter [7,9]”

•7) Why nanoparticles 1 and 2 are presented in the table S1, while they are neither shown nor discussed in the manuscript. What are the effects of the previous cycles on the studied NP (NP300 and NP650)?

Answer: The purpose of Table S1 was to give to the reader a complete history of the gas cycles to put in perspective the measurements carried out on NP300 and NP650.

NP1 was measured at the beginning of the experiment. It is a particle containing a dislocation. While these kind of NPs are interesting to study, the extended displacement field induced by the dislocation tends to mask or at least make more difficult to appreciate the changes taking place on the NP surface. We therefore decided not to include this data in the manuscript. We added a Figure in the Supplementary Materials (Supplementary Figure S9) showing DPs and the corresponding reconstructions in Ar and Ar + O₂ (2%) atmosphere.

Regarding NP2, it is also clear from the DP that this NP was defective, unfortunately the phase retrieval did not converge and is therefore included in the present work.

Is there any ageing effect on the catalytic material? Please comment on possible effects on the strain of 19h hours separating the two cycles investigated by BCDI for NP650.

As discussed in the manuscript and in the answer to Q2 of referee n°1, there is no apparent ageing effect for NP650. The displacement (Fig. S6) and strain fields (Fig. S8) are very similar between condition (a) (S787) and condition (h) (S1244) despite the fact that NP650 experienced 3 reducing and 3 stoichiometric CO oxidation during the 19 hours that separate the two measurements. In addition, the surface strain states in both condition (a) and condition (h) are very similar to the simulation results, suggesting that in both cases the surface of NP650 is mostly free of adsorbates.

For NP300 on the other hand the initial surface strain state is not fully recovered after the first stoichiometric CO oxidation (SC4), suggesting a possible ageing effect.

•8) Although obtained in different experiments, it would be interesting to show the evolution of catalytic performances of the sample during different cycles. Indeed, the type/number of cycles might directly alter the strain distributions observed within the BCDI study.

Answer: We added in the Supplementary Materials the mass spectrometry data collected on a similar sample during CO oxidation (Supplementary Materials S22). The catalytic activity is the highest in stoichiometric conditions compared to reducing conditions, however the effect of cycling was not explored during this experiment.

Based on strain observations, it appears that the activity is the highest during SC4 for NP300 while the activity remains more or less constant throughout the experiment for NP650 (reversible surface strain state at the beginning of each cycle).

•9) Are the catalytic performances affected in the same way between the stoichiometric and non-stoichiometric conditions for both sizes of NP? Can it be correlated to the differences observed on the displacement field and strain distributions?

Answer: This is discussed in quite extensive details in the paper. For NP650 the strain evolution is very similar in stoichiometric conditions and in reducing conditions: *i. e.* overall identical facets show equivalent catalytic response. We rationalised this behavior in terms of oxygen adsorption or desorption during O₂ exposure or CO oxidation in reducing conditions. For NP300, the facet-dependent strain evolution is also observed during reducing CO oxidation. During stoichiometric CO oxidation (SC4) on the other hand, large strain variations are observed, in particular close to the NP/substrate interface. These variations are no longer facet-dependent. Plodinec *et al.* suggested that these reaction induced structural changes are characteristic of a high activity regime [10]. This would indicate that the highest activity is obtained in stoichiometric conditions for NP300, while there is not much difference between the stoichiometric and reducing conditions for NP650.

•10) Did the authors investigate "fresh" samples (without previous cycles)? What did they observe?

Answer: We indeed measured a NP, NP1 in its pristine “fresh” state. However, as discussed previously, this particle contains a dislocation whose large and extended displacement field tends to mask the displacement / strain evolution occurring during adsorption and reaction. Nonetheless, the strain evolution during oxygen exposure and stoichiometric and reducing CO oxidation appears to be consistent with the O₂ adsorption / desorption mechanism we described in the manuscript (strain relaxation during oxygen exposure followed by an increase of the surface strain during stoichiometric and non-stoichiometric CO oxidation) for this NP. In addition, the fact that we manage to obtain the same “adsorbate free” initial strain state in condition (a) and (h) demonstrates that the ageing effect is very limited for NP650 (but not in NP300 where the initial strain state is not completely recovered).

•11) Regarding the MS simulations, the authors simulated NP of ~ 32 and 200 nm. Please provide details about the 32 nm simulations. Also why choosing a 200 nm NP for the simulation? Please comment on the difference of sizes between the simulations and experiments. Does the size affect the average strain obtained on the simulated NP? Please add a new figure for NP300 with the comparison between experiment and simulation (like figure 4 but for NP300).

Answer: We have not just simulated NP of 32 and 200nm but considered a range of size (6 different sizes) between these two extrema in order to evaluate/quantify the size effect on the strain/displacement distribution.

We could draw two conclusions from this study: (1) the size effect on the displacement is very limited, and (2) a very similar displacement distribution is obtained for the smallest simulated NP (32 nm) and the largest (200 nm). The strain being the derivative of the displacement, the bulk strain obviously scales with the size of the NP (the smaller the NP, the larger the bulk strain). The very large strain on the topmost surface layers is on the other hand mostly independent on the size of the NP, meaning that the surface strain field largely depends on the choice of the voxel size (number of layers over which the strain is averaged). This emphasizes the need to use a voxel size similar to the experimental one in order to achieve a quantitative comparison between experiment and simulation. That’s why, simulating a smaller NP with the same voxel size as the experiment gives a good agreement between experiment and simulation.

We added a figure showing the simulated NP300, the calculated strain field is also in very good agreement with the one we measured experimentally (Supplementary Figure S23).

•12) Did the author observe the formation of an interfacial dislocation network, as mentioned page 7? Please comment and identify those dislocations.

Answer: Unfortunately, we did not manage to collect any Transmission Electron Microscopy image to characterize the NP / substrate interface and establish unambiguously the presence of a network of interfacial dislocations as well as the type of these dislocations. The formation of an interfacial dislocation is one of the preferred deformation mechanisms in order to relax large elastic strains and is commonly observed in NPs in epitaxy with a substrate during thermal treatments [11].

In this experiment, we couldn’t capture the formation of the interfacial dislocation network, since we started to measure NP650 and NP300 only after reaching 450°C.

We assumed the presence of interfacial dislocations for several reasons:

- 1) Purely elastic displacement fields are smooth and continuous, while crystal defects such as dislocation induce sharp variations in the phase (phase jumps, such as the one we observe at the interface)
- 2) The presence of such dislocations has been reported previously in the literature in Au NPs following a similar preparation route on the same (0 0 0 1) sapphire substrate
- 3) We observed the formation of such dislocation network during the heating of Ni NPs on a sapphire substrate.

•13) Overall, there are too many references. Their number should be reduced, with a better selection. Please check typing for the following references: 11; 13; 26; 29; 31; 35; 38; 50; 51.

Answer: We have reduced the number of references and corrected the typos.

•14) Although the whole figures cannot be presented in the main manuscript, I would suggest showing only the ones concerning NP300 in the main manuscript and the ones of NP650 in supplementary. Mixing both NP in the figures of the manuscript/supplementary is too confusing.

Answer: We believe that one of the strength of this paper is to demonstrate that two NPs with different sizes and surface morphologies (faceting) can exhibit a different behavior during stoichiometric CO oxidation. For this reason we prefer to keep the organisation of the figures as it is in order to emphasize the similarities and difference of strain evolution during the reaction. In addition, we believe that the captions of the figures are explicit enough in order to prevent any confusion.

•15) Figure 3,b, please modify the symbols used for $\langle \epsilon 111 \rangle$ from diffraction patterns and from reconstruction, we can hardly distinguish them.

Answer: This is now done.

•16) Figure 4. What is shown in a/c and b/d? the same simulated NP and reconstructed NP with different field of view? Or beginning / end of the gas mixture? Please modify the figure caption.

Answer: These are the same NP but from seen from different fields of view (a,c: Simulated NP650; b,d: Experimental NP650). We modified the figure caption according to the reviewer suggestion.

•17) What are the corresponding conditions of Figure S8?

Answer: Fig S12 represents NP650 in 100% Ar. This is the first measurement carried out on this particle corresponding to our reference state (S787). This is now clearly indicated in the caption of the figure.

•18) In the conclusion: "our operando observation of how an individual nanoparticle "breathes" during CO oxidation reaction is unprecedented in its picometer-scale displacement resolution"

This work is far from being an operando study. As described in Nature Reviews Materials, 3 (2018) 324: "operando means studying the catalyst under the true reaction conditions, while in situ refers to both model and true reaction conditions. An operando study also requires the simultaneous online analysis of catalyst performance (activity and/or selectivity) using, for example, mass spectrometry (MS) or gas chromatography (GC) to measure product concentrations."

Thus, this study can be considered as an in situ study, but due to the lack of online analysis of catalyst performance, this is not an operando study.

Please comment and modify the conclusion.

Answer: We agree with the reviewer and modified the conclusion accordingly.

Reviewer #3 (Remarks to the Author):

1) The manuscript reports on the study of strain in faceted Pt nanocrystals under controlled reaction conditions using Bragg coherent diffraction imaging, which has been developed over the years into a quantitative method for imaging 3D strain at the spatial resolution tens of nm as the cited references show (Refs. 38-44). This work concerns the development of strain and its facet dependence. Two nanocrystals of size 350 and 600 nm were investigated over multiple diffraction scans and a combination of gas reaction conditions.

The major findings are related to the observation of strain and displacement fields and dependence on reaction conditions. The discussions on size-dependence are less convincing the sampling regarding particle size is very limited and the size of the measured particles generally are of hundreds of nm, which are large for nanoparticles. An extension of the study to smaller nanoparticles could be helpful, but it is not clear what the smallest nanoparticles that could be investigated by this technique.

Answer: We thank the reviewer for this evaluation of the manuscript and understand his concerns regarding the size of the investigated NPs. The sampling might appear indeed very limited but this is obviously related to the time consuming character of these measurements. As illustrated in Table S1, performing a complete set of measurements on a single NP requires at least 24h, making impractical the measurement of a large number of NP.

Regarding the minimum size of the measured NPs, state of the art literature using the technique reported measurements carried out on 60 nm [12] which is still significantly larger than the NPs size used in industrial catalysts (typically below 10 nm). With the advent of fourth generation sources (EBS upgrade of the ESRF for instance), it should be possible to measure smaller NPs (of the order of 20 nm), but it might prove challenging to perform these measurements *in situ*.

2) The strain and displacement fields are presented in figs. 2, 4 and 5 as three-dimensional surface color plots. Are these the measured surface strain and displacements?

Answer: The strain and displacement fields presented in Figs. 2, 4 are indeed the measured 3D displacement and strain fields averaged over the voxel size. The latter depends on the size of the \mathbf{q} range over which the measurement is carried out in the reciprocal space: $d = 2\pi/\mathbf{q}$, and should not be confused with the spatial resolution (which is typically slightly worse). In this work, the voxel size (corresponding to an average of the voxel size along the 3 directions) is equal to 6.7 nm. This implies that the surface strain is actually averaged over the 30 topmost surface layers. This is why the values of the surface strain reported in this work and obtained from simulation are relatively low (few 10^{-4}). The strain on the topmost layer is actually significantly larger (typically of the order of 10^{-2} depending on the crystallographic index of the facet). In order to get a quantitative agreement between the simulation and the experiment, the simulated voxel size must match the experimental one.

What does "drawn at 50% of ..." mean in fig. 2 caption?

Answer: BCDI allows to reconstruct a complex image of the nanocrystal from a diffracted intensity. Its amplitude (modulus) corresponding to the Bragg electron density while its phase encodes the projection of the displacement field $\mathbf{u}(\mathbf{r})$ onto the scattering vector \mathbf{g} . For a more detailed description of the amplitude and phase terms, please refer to answer of Q5 from referee n°1 and Refs [3,4].

For a monoatomic material, the Bragg electron density (we add the term Bragg because its value depends on the scattering vector considered) should be flat and have the same value everywhere in the crystal. Basically if we normalize this value, it should be equal to one for all the voxels belonging to the crystal and zero outside.

However, because of the inherent noise of the experimental measurements, the imperfections of the beam, or the partial illumination of the measured crystal, the measured Bragg electron density might fluctuate in the nanocrystal (again this is not physical, this is an artifact of the measurement). This is now illustrated in Fig. S13 which shows the distribution of the electron density in NP300 in Ar atmosphere, together with a histogram showing the distribution of the normalized amplitudes in the reconstructed data. As seen from Fig. S13, the voxel belonging to the NP have roughly a Gaussian distribution centered around 0.72 with a standard deviation of 0.19. This value of 53% corresponds to the mean of this Gaussian distribution subtracted by the standard deviation. For more information on the importance of the choice of a good isosurface and its impact on the evaluation of the surface strain please refer to Ref. [13].

3) The Pt nanoparticles were formed by annealing in air at 1100 C. Why in air? Does Pt oxidize under such condition? Is heating at 450 C under argon sufficient to drive out oxygen for a comparison with MD calculation (the agreement between theory and exp is remarkably good, could this indicate an insensitivity of the measured strain to actual surface structure?).

Answer: Our goal was to obtain clean faceted Pt particles with a limited number of crystallographically well-defined facets. This is a formidable task since it is known that even annealing in ultrahigh vacuum (UHV) of 5×10^{-10} Torr at the temperature of 1200°C is not sufficient in getting rid of carbon contamination on the Pt particles [13]. Carbon impurities can only be eliminated by subsequent annealing in oxygen containing ambient which removes carbon in the form of CO or CO₂ molecules and purifies the particles. However, the resulting shape of the Pt particles is nearly spherical, with only minor {1 1 1} and {1 0 0} facets [14]. Such spherical particles expose the full range of surface orientations and they are not suitable for the studies of the effect of environment on the stress state of specific facets.

On the contrary, we found that annealing the Pt film at the temperature of 1100°C in air results in clean particles of a polygonal shape with a limited set of crystallographically well-defined facets [15]. The oxidation of Pt is not an issue since above approximately 1000°C Pt forms a volatile oxide PtO₂ which readily evaporates. The mass loss of Pt during annealing in air has already been noticed by Edison at

the end of 19th century [16]. The evaporation of PtO₂ plays an additional positive role since it eliminates contaminated surface layers of the film and of the particles.

4) The surface referred here presumably has a thickness, rather than single or few atomic layers that are studied by microscopy techniques. The measured strain is for the 111 lattice plane and thus does not provide information the full strain fields. What is measured and presented should be made clearer to the readers.

The selection of (111) reflection rather than (200) or (113) is likely coming out the convenience of the experiment. Whether such condition is suitable for the study of surface strain in a nanoparticle with different facets should be discussed.

Answer: As mentioned in a previous answer, the surface strain and displacement fields are indeed averaged over a thickness corresponding to the voxel size (6.7 nm or 30 (1 1 1) atomic layers).

The selection of a 1 1 1 reflection was indeed dictated by experimental convenience and time constraints. As mentioned in the methods section, the NPs are in epitaxy on the sapphire substrate and have a [1 1 1] out-of-plane orientation, allowing to carry out the measurement in specular geometry. In addition, as seen from Table S1, the measurements are extremely time consuming, making them difficult (but not impossible, we performed such experiment recently) to measure several reflections during an *in situ* experiment.

It is correct that measuring a single Bragg reflection as we did during this experiment is not enough to measure the full 3D strain field. However it already gives valuable information on the strain evolution in the presence of gas adsorbates or during reaction.

Moreover, as already pointed out in the answer to Q3 from referee n°2, the relaxation of the surface atoms does not occur only perpendicular to the facet. For the {1 -1 0} and {1 1 -3} facets (the selected **q** vector is basically insensitive to the displacement perpendicular to these facets (out-of-plane displacements)), however, it is very sensitive to large in-plane displacements (parallel to the facet) in the vicinity of these facets.

Based on these considerations, we believe (and the results presented in this manuscript speak for themselves) that the measurement of a single Bragg reflection is suitable for the study of surface strain in a NP with different facets. That being said, we also agree that measuring 3 non-coplanar Bragg reflections would give a more complete picture of the strain evolution during reaction and will be object of a future work/publication.

→ We added a sentence in the beginning of the results section, emphasizing the fact that the measurement of a single Bragg reflection is not sufficient to obtain the 3D strain field.

References

- [1] Sayre, D. *Some Implications of a theorem due to Shannon*. Acta Cryst. 5, **843** (1952).
- [2] Li, W. X. et al. *Oxidation of Pt(110)*. Phys. Rev. Lett. **93**, 146104 (2004).
- [3] Robinson, I. & Harder, R. *Coherent X-ray diffraction imaging of strain at the nanoscale*. Nat. Mater. **8**, 291–298 (2009).
- [4] Clark, J et al. *Ultrafast Three-Dimensional Imaging of Lattice Dynamics in Individual Gold Nanocrystals*. Science **341**, 56 (2013).
- [5] Carnis J. et al. *Twin boundary migration in an individual platinum nanocrystal during catalytic CO oxidation*. Nature. Comm **12**, 5385 (2021).
- [6] Tran, R., Xu, Z., Radhakrishnan, B., Winston, D. W. Sun, K. A. Persson, S. P. Ong, *Surface Energies of Elemental Crystals*, Scientific Data **3**, (2016).
- [7] Tran, R., Li, X-G., Montoya, J. H., Winston, D., Persson, K. A., Ong, S. P. *Anisotropic Work Function of Elemental Crystals*, Surface Science **687**, (2019).
- [8] Zeng, H., Li, X-G., Tran, R., Chen, C., Horton, M., Winston, D., Persson, K. A., Ong, S. P. *Grain Boundary Properties of Elemental Metals*, arXiv:1907.08905 (2019).
- [9] Kim, D., Chung, M., Carnis, J. et al. *Active site localization of methane oxidation on Pt nanocrystals*. Nat Commun **9**, 3422 (2018).

- [10] Plodinec, M., Nerl, H. C., Girgsdies, F., Schlögl, R. & Lunkenbein, T. *Insights into Chemical Dynamics and Their Impact on the Reactivity of Pt Nanoparticles during CO Oxidation by Operando TEM*. ACS Catalysis **10**, 3183–3193 (2020).
- [11] Sadan H. & Kaplan W. D. *Au–Sapphire (0001) solid–solid interfacial energy*. J. Mater. Sci. **41**, 5099–5107 (2006).
- [12] Björling, A. *et al.* *Coherent Bragg imaging of 60 nm Au nanoparticles under electrochemical control at the NanoMAX beamline*. J. Synchrotron Rad. **26**, 1830–1834 (2019).
- [13] Carnis, J. *et al.* *Towards a quantitative determination of strain in Bragg Coherent X-ray Diffraction Imaging: artefacts and sign convention in reconstructions*, Sci. Rep. **9**, 17357 (2019).
- [14] Lee, WH, Vanloon, KR, Petrova, V, *et al.* *The equilibrium shape and surface energy anisotropy of clean platinum*. J. Catal. **126**, 658–670 (1990).
- [15] Zimmerman, J, Bisht, A, Mishin, Y, Rabkin, E. *Size and shape effects on the strength of platinum nanoparticle*. J. Mater. Sci. (2021).
- [16] Edison, TA, *Heating Metals in Vacuum by the Electric Current*. Am. J. Dent. Sci. **13**, 418 (1880).

REVIEWER COMMENTS

Reviewer #1 (Remarks to the Author):

Even though they tried to answer all the questions, I am not yet convinced whether the quality of this paper satisfies the standard of Nature Communications for the following reasons. I recommend it to publish in a nanoscience journal.

1. It is still too difficult for the readers to follow their presentations based on their experimental logbook (scan numbers, measured time, and sample naming, etc.) shown in S1 in the Supporting Information.
2. As they confirmed in their reply, their observations were done only one experiment from the samples on one substrate. It means that the samples had all different histories with thermal and gas exposures when they were measured. They showed several measurements of the different particles at “the same condition” in the beginning and at the end (arbitrary time), but different histories with gas exposure. At least if they want to compare the “size” dependence or different kinds of cycles, the starting points of the sample should be measured.
3. For their reply to explain the displacement and strain, what I originally meant was how they obtained surface displacements and surface strain instead of the principle of the techniques. Since the “surface” can be determined by the isosurface of the results, as they mentioned in reply to Referee #3, they are averaged over a thickness to the voxel size.

Reviewer #2 (Remarks to the Author):

The authors answered most of my comments and questions, however the new Figure S2 b, showing SEM images of the Pt sample, confirms my concerns about considering this sample as relevant for catalysis. Indeed, although this beautiful mis-titled work reports interesting In situ imaging of the facet surface strain state on multi-faceted Pt nanoparticles (not catalytic reaction), I have strong reservation from the catalytic point of view.

The 300 and 650 nm Pt nanoparticles are far too large for ideal catalytic performance. It is well established that the size (and size distribution) of the nanoparticles can directly affect the catalytic properties of a material. Working with an inhomogeneous sample can lead to inaccurate conclusions. In this work, what is the size distribution of the particles in the reactor? From Fig. S2, it is obvious that the sample is presenting a very broad size distribution, which is far from ideal for the catalysis point of view.

Did the authors quantify the size distribution? Was the reaction actually activated by smaller Pt particles that reside on the support, rather than the studied ones by Bragg-CDI?

How does inhomogeneous particle size affect the properties (structural, lattice distortion, catalytic properties...) of the “catalytic” material? What’s the effect of the smallest particles? Could the

differences observed between both sizes also come from the fact that NP300 is participating to the reaction, while NP650 is not participating (or less active)?

In summary, considering the new information provided by Fig. S2, leading to my main concern, the manuscript should be at least revised. Furthermore, the title needs to be changed accordingly. It is highly misleading to imply the studied Pt nanoparticles are responsible of the catalytic reaction, considering that much smaller nanoparticles are present on the studied sample.

Moreover, the following concerns should be addressed:

- All the figures/tables presented in Supplemental Materials should be cited once (at least), either in the main manuscript or Supplementary materials (for example, S6 (with figure S5) does not appear). Please, check other supplementary figures/tables/notes.
- The general organization of Supplemental Materials should be checked as well. The order of the figures/tables should be consistent with the manuscript.
- New information and figures provided in the answers for referees should be included into the main manuscript or Supplemental materials. They are too many points discussed in the answers but not implemented in the article (e.g. what are NP1 and NP2?, time/conditions of acquisition of Bragg CDI scans...).
- Figure 3 is presenting results for NP300. For a better understanding of the reader, the same kind of figure should be added for NP650 (in Supplemental materials?). In particular, how is the strain field energy affected for NP650? What about the micro strain evolution?
- Figure 3: Why is the strain field energy with so few points? Are they average points? If yes, put min/max values. Or why did the authors choose to present only few points?
- Figure 6: Is it an average for all Bragg-CDI data of each condition? or with the first/last data? What are the differences between the first and last data?
Please clarify it.
Same question for Fig. S14 and S15, and tables with similar information...
- S1:
 - Since it is also part of the history of the sample, the step of heating under Ar should be included in this table.
 - why SC3 is written two times and referencing to different gas percentages? Should they be considered as two different cycles?
 - it might be good to add a column with the number of Bragg CDI scans done for each condition (this is not obvious with the scan number presented here)
 - add in the table the letters referencing to the different conditions used in Fig. 2 and Fig. S6.

- clarify in the text of S1 what are NP1 and NP2.
 - in the text of S1, describe the time/conditions for the scans. If not mentioned in S1, should be added into Methods.
 - line S842-S856, RC1 and 10 are not in the good columns
- S2:
 - In the caption, add the gas used during the heating ramp.
 - What did the author observe with the scans measured during the heating ramp? How the DP are affected by the increasing temperature? The scans have been obtained for the same NP? Same size? 300 nm? 650 nm? Other?
- S3:
 - S3 is not mentioned, neither in the manuscript nor in Supplemental Materials. However, it needs to be included and better discussed considering the rather wide size distribution, that cannot be unheeded in the discussion.
 - To know if NP300 and NP650 are representative of the sample, (as the authors suggested), the authors need to quantify it, for example, by adding a histogram obtained with size measurements of the NP from electron microscopy images (average size, dispersion...)
- S13:
 - Modify Start / End which is too confusing.
- S23:
 - It seems that this figure is not showing the conversion of the sample presented in this paper. This should be clearly described in the figure caption.
 - Moreover, S23 does not appear in the main manuscript (and is not referred in the Supplemental Materials). As already mentioned, a figure presented should be discussed, otherwise it should not be presented.
- Fig, 2 (and equivalent in SM):
 - Add first/last measurement on the image itself. It would help the reader to understand (faster) what is shown

Other minor corrections in the pdf file.

Reviewer #3 (Remarks to the Author):

The authors' response and revision of manuscript addressed my main concern about the measurement nature, which samples strain along a particular direction, e.g, (111), in 3D. In authors' reply, the limitation of using the authors' technique for studying smaller nanocrystals was discussed. While this

limits the significance about size-dependence, the report represents the state of art and could motivate further experiment on this topic. Publication is recommended.

REVIEWER COMMENTS

Reviewer #1 (Remarks to the Author):

Even though they tried to answer all the questions, I am not yet convinced whether the quality of this paper satisfies the standard of Nature Communications for the following reasons. I recommend it to publish in a nanoscience journal.

1. It is still too difficult for the readers to follow their presentations based on their experimental logbook (scan numbers, measured time, and sample naming, etc.) shown in S1 in the Supporting Information.

Answer: We have made several changes in order to improve the readability of the manuscript and Table S1:

- We have modified the labeling in Table S1. Particles 1 and 2 are now referred as NPD1 and NPD2. The former is also referred as NPD1 in Supplementary Fig. S2. Particle 3 (previously mentioned in Table S1) corresponds to NP650. We agree that it was not completely clear in the previous version of Table S1 and we fix this issue accordingly.

- We have also added a column named T(°C) in Table S1 to show the temperature at which each measurement was carried out. We have also added the number of BCDI scans and total time for a given gas condition.

- Each line of the table is now associated to a single nanoparticle, either NPD1, NPD2, NP300 or NP650.

- We have added a second simplified table (Table S2) to show only the measurement carried out on NP300 and NP650, which are the two particles presented in the manuscript.

- Finally we modified the order of the Figures in the Supplementary Materials and we made sure that they are cited at least once in the manuscript or Supplementary Materials.

2. As they confirmed in their reply, their observations were done only one experiment from the samples on one substrate. It means that the samples had all different histories with thermal and gas exposures when they were measured. They showed several measurements of the different particles at “the same condition” in the beginning and at the end (arbitrary time), but different histories with gas exposure. At least if they want to compare the “size” dependence or different kinds of cycles, the starting points of the sample should be measured.

Answer: As discussed in the manuscript, we have measured two particles (NP300 and NP650) on the same substrate. It was not possible to measure the particles simultaneously. We have measured the particles one after the other. Note that the two particles show the same thermal history. Before measurement, the two particles have been reduced under CO, so that the surface of the particles should be free of O₂ adsorbates in the initial state. For the two particles, the starting point is a reduced Pt particle at 450°C.

3. For their reply to explain the displacement and strain, what I originally meant was how they obtained surface displacements and surface strain instead of the principle of the techniques. Since the “surface” can be determined by the isosurface of the results, as they mentioned in reply to Referee #3, they are averaged over a thickness to the voxel size.

Answer: The surface strain is averaged over a thickness corresponding to the voxel size. This was discussed in extensive details in our previous rebuttal (answer to Q2 and Q11 as well as answer to Q2 and Q4 of reviewer #3). This is now clearly written in the main manuscript: “The voxel size (corresponding to an average of the voxel size along the 3 directions) is equal to 6.7 nm. This implies that the surface strain is actually averaged over the 30 topmost surface layers.” The dependence of the surface strain to the selection of the isosurface was also discussed in the previous rebuttal (answer to Q2 of reviewer #3). We have also added a sentence in the new version of the manuscript to discuss the influence of the choice of the isosurface on the surface strain distribution: “The surface strain also depends on the selection of the isosurface on the experimental data, as discussed in Fig. S6 and in Ref. [Carnis *et al*, *Sci. Rep.* 9, 1–13 (2019)]”. In Fig. S6, we have added: “The voxels belonging to the NP have roughly a Gaussian distribution centered around 0.72 with a standard deviation of 0.19. This value of 53% corresponds to the average of this Gaussian distribution subtracted by its standard deviation, and it is a good estimate for the choice of the isosurface on the surface strain distribution.”

Reviewer #2 (Remarks to the Author):

1. The authors answered most of my comments and questions, however the new Figure S2 b, showing SEM images of the Pt sample, confirms my concerns about considering this sample as relevant for catalysis.

Indeed, although this beautiful mis-titled work reports interesting In situ imaging of the facet surface strain state on multi-faceted Pt nanoparticles (not catalytic reaction), I have strong reservation from the catalytic point of view. The 300 and 650 nm Pt nanoparticles are far too large for ideal catalytic performance. It is well established that the size (and size distribution) of the nanoparticles can directly affect the catalytic properties of a material. Working with an inhomogeneous sample can lead to inaccurate conclusions.

Answer: We have removed the word “nanocatalysts” in the title of the manuscript and replaced it with “nanoparticles”. Even if the sizes of the particles are larger than the ones used for industrial applications, the particles studied in this work can be considered as a model system for catalysis. Given the size distribution of the NPs used in this work, this study is of course not aiming at achieving the best catalytic performance. The choice of large NPs for this study is dictated by the spatial resolution of the BCDI technique (~ 10 nm). Nonetheless, we strongly believe that the observations presented in this manuscript are very valuable to get a better understanding between strain and activity for several reasons:

1) These large NPs do present catalytic activity at high temperature as demonstrated in Fig. S24 (mass spectrometry from a different sample with Pt particles very similar in size to the sample measured in the manuscript).

2) Although the NP size in practical uses is a few nanometers, we believe that the local atomic displacement due to the interaction of reactants to the catalyst can be understood even in the size range used in this study (~300 - 650 nm). More importantly, this study indeed adds to our understanding of the bulk and surface strain evolution resulting from the adsorption of gas species or induced by reactions taking place on the NP surface and their dependence of the facet type. For instance we revealed the dependence of the facet type for the oxygen adsorption and desorption mechanisms taking place during O₂ exposure and during reducing CO oxidation, respectively.

These changes cannot be probed by any other technique and justify in our opinion the use of BCDI to investigate the relationship between strain and activity. We are convinced that these findings can be applied to smaller nanometer-scale particles.

3) Thanks to the upgrades of synchrotron sources, the spatial resolution of the technique is likely to improve dramatically in the future. This will allow to investigate catalyst sizes that are more relevant for industrial applications.

2. In this work, what is the size distribution of the particles in the reactor? From Fig. S2, it is obvious that the sample is presenting a very broad size distribution, which is far from ideal for the catalysis point of view.

Did the authors quantify the size distribution? Was the reaction actually activated by smaller Pt particles that reside on the support, rather than the studied ones by Bragg-CDI?

Answer: The particle size ranges from 100 nm to 1 μm. We have added an histogram in Figure S1, showing the average size and size distribution of the sample extracted from the SEM picture shown in Fig. S1c (average size: (550 ± 230) nm).

It is indeed very likely that the smaller particles are more active towards the reaction. However, it would be a very difficult task to determine if only small particles (< 100 nm in size for instance) contribute to the reaction since it is not possible to measure the activity of a single particle with our setup.

In any case, it is very probable that the inhomogeneous particle size will affect the overall activity of the sample (ensemble average properties). Nonetheless, we believe that the changes in the structure and strain (local properties) observed during oxygen adsorption or CO oxidation in NP300 and NP650 can be representative of the strain changes experienced by smaller NPs during the reaction. The smallest NPs should contribute more to the reaction and it is very likely that the large lattice distortions observed in NP300 during stoichiometric CO oxidation also take place in smaller NPs.

3. How does inhomogeneous particle size affect the properties (structural, lattice distortion, catalytic properties...) of the "catalytic" material? What's the effect of the smallest particles? Could the differences observed between both sizes also come from the fact that NP300 is participating to the reaction, while NP650 is not participating (or less active)?

Answer: NP300, which is significantly smaller than the average NP size on the sample, experiences by far its largest strain evolution during stoichiometric CO oxidation. This clearly suggests that this NP is active during the reaction. This would be indeed consistent with the observations of Plodinec *et al.* which reported that high activity regimes are associated to large lattice distortions.

As mentioned previously, the smallest NPs should contribute more to the reaction and it is very likely that the large lattice distortions observed in NP300 during stoichiometric CO oxidation also take place in smaller NPs.

In contrast, the changes are less pronounced in NP650. This indeed points to a lower activity of this NP towards the reaction. As discussed in the manuscript, it is not entirely clear at this stage if the different behavior between NP650 and NP300 is caused by the difference in size (650 nm vs 300 nm) or by the different faceting (larger proportion of {1 1 3} facets for NP650) and would require further investigation.

4. In summary, considering the new information provided by Fig. S2, leading to my main concern, the manuscript should be at least revised. Furthermore, the title needs to be changed accordingly. It is highly misleading to imply the studied Pt nanoparticles are responsible of the catalytic reaction, considering that much smaller nanoparticles are present on the studied sample.

Answer: As mentioned previously, we agree that their size is larger than the optimum size range, where Platinum is considered to be the most catalytically active: we therefore replaced the word “nanocatalyst” by “nanoparticles” in the title of the manuscript.

However, we disagree with the statement that it is highly misleading to imply that the studied Pt nanoparticles are responsible of the catalytic reaction for several reasons that were discussed in our previous answers:

- 1) NP300 is smaller than the average NP size on the sample,
- 2) NP300 experiences by far the largest structural/strain evolution during stoichiometric CO oxidation. These large distortions (increase of the FWHM of the Bragg peak) are associated to high activity regimes as suggested by Plodinec *et al.*
- 3) We also observe reversible and clear changes in both NP650 or NP300 during the cycles performed in reducing conditions.

5. Moreover, the following concerns should be addressed:

- All the figures/tables presented in Supplemental Materials should be cited once (at least), either in the main manuscript or Supplementary materials (for example, S6 (with figure S5) does not appear). Please, check other supplementary figures/tables/notes.

Answer: All the figures/tables/notes are now cited at least once either in the main manuscript or in the Supplementary Materials.

- The general organization of Supplemental Materials should be checked as well. The order of the figures/tables should be consistent with the manuscript.

Answer: We have checked the organization of the Supplemental Materials such that it is consistent with the manuscript.

- New information and figures provided in the answers for referees should be included into the main manuscript or Supplemental materials. They are too many points discussed in the answers but not implemented in the article (e.g. what are NP1 and NP2?, time/conditions of acquisition of Bragg CDI scans...).

Answer: We have added additional details regarding the time/conditions of acquisition of Bragg CDI scans in the methods section and in Supplementary Tables S1 and S2.

We are also now providing additional details on the measurements carried out on the first cycles on NPD1 (Supplementary Tables S1 and S2 and Supplementary Figure S2) and NPD2 (Supplementary Tables S1 and S2).

- Figure 3 is presenting results for NP300. For a better understanding of the reader, the same kind of figure should be added for NP650 (in Supplemental materials?). In particular, how is the strain field energy affected for NP650? What about the micro strain evolution?

Answer: We inserted the same figure for NP650 in Supplementary Materials. The strain field energy is higher for NP650 compared to NP300. This is maybe due to the larger size of the particle. No clear variation of the micro-strain is observed for NP650.

- Figure 3: Why is the strain field energy with so few points? Are they average points? If yes, put min/max values. Or why did the authors choose to present only few points?

Answer: In the previous version of the manuscript, we have chosen to show only few points (1 point per gas condition). In the new version of the manuscript, we added all the points corresponding to the conditions shown in Fig. 2.

- Figure 6: Is it an average for all Bragg-CDI data of each condition? or with the first/last data? What are the differences between the first and last data? Please clarify it.

Answer: Fig. 6 describes the average strain evolution ($\Delta\epsilon_{111}$) per $\{hkl\}$ facet family in NP300 and NP650. We have chosen to represent the strain variation between two consecutive gas conditions to have a clearer picture of the strain dynamics.

For instance the $\{1 -1 1\}$ facets, that are initially in compression, show a positive strain variation ($(\Delta\langle\epsilon_{111}\rangle > 0)$) when switching from Ar to Ar + 2.5% O₂ atmosphere, which means that these facets experience a relaxation of the compressive strain in oxygen atmosphere. Similarly, when

switching from Ar + 2.5% O₂ atmosphere to reducing oxidation conditions, ($\Delta\langle\epsilon_{111}\rangle < 0$) meaning that these facets resume their initial compressive strain.

The opposite trend is observed for the other facet families: *i.e.* a relaxation of the tensile strain in O₂ atmosphere ($(\Delta\epsilon_{111}) < 0$) followed by an increase in reaction conditions ($(\Delta\epsilon_{111}) > 0$).

We slightly modified the text and the caption so that the interpretation of this figure is easier for the reader.

Same question for Fig. S14 and S15, and tables with similar information...

Answer: Figs. S14 and S15 (now Figs. S16 and S17) provide a similar information except this time, all the facets belonging to each family are presented one by one. This allows to confirm that the facets belonging to the same family experience similar strain dynamics during a gas cycle and demonstrate that the strain evolution is mostly facet dependent in reducing conditions.

S1: - Since it is also part of the history of the sample, the step of heating under Ar should be included in this table.

Answer: We added the scans corresponding to the heating of the sample in Table S1 and S2 as well as a column showing the temperature at which the measurements have been performed.

- why SC3 is written two times and referencing to different gas percentages? Should they be considered as two different cycles?

Answer: These conditions were both referred as SC3 because they were measured sequentially and with the same ratio between the reactants ($\chi = 2$). The second SC3 is now labeled SC3' since the fraction of CO and O₂ is doubled in the gas mixture. This is now explicitly written in Table S1.

- it might be good to add a column with the number of Bragg CDI scans done for each condition (this is not obvious with the scan number presented here)

Answer: We have added a column with the number of BCDI scans done for each condition in Table S1.

- add in the table the letters referencing to the different conditions used in Fig. 2 and Fig. S6.
- clarify in the text of S1 what are NP1 and NP2.

Answer: We have added in new Tables S1 and S2 the letters referencing to the conditions used in Figs. 2 and S7-S9.

We apologize for the inconsistent labeling for NP1 and NP2 and thank the reviewer for pointing it out. They are now always referred as NPD1 and NPD2 throughout the manuscript.

- in the text of S1, describe the time/conditions for the scans. If not mentioned in S1, should be added into Methods.

Answer: We have added this information in the Methods section.

- line S842->S856, RC1 and 10 are not in the good columns

Answer: We thank the reviewer for pointing out this typo and we made the appropriate corrections.

• S2:

- In the caption, add the gas used during the heating ramp.

Answer: We added this information in the caption of the figure (now Fig. S3).

- What did the author observe with the scans measured during the heating ramp? How the DP are affected by the increasing temperature? The scans have been obtained for the same NP? Same size? 300 nm? 650 nm? Other?

Answer: The thermoelastic strain field at the interface with the substrate is of course dependent of the temperature. Generally we observe a relaxation of this elastic strain field during the heating ramp, but the situation might be different if the elastic strain field was relaxed at room temperature by a network of interfacial dislocations. This is a very interesting topic that will be the object of a future publication but which is not in the scope of this study.

We did not measure NP300 and NP650 during the heating ramp and are therefore not able to comment on the strain evolution from RT to the reaction temperature (450°C). Nonetheless, the inhomogeneity of the strain distribution at the interface suggests the formation of a network of interfacial dislocations during heating in both particles.

•S3:

- S3 is not mentioned, neither in the manuscript nor in Supplemental Materials. However, it needs to be included and better discussed considering the rather wide size distribution, that cannot be unheeded in the discussion.

Answer: Fig. S3 (now called Fig. S1) is now mentioned in the manuscript. We have added an histogram of the size distribution of the NPs to put in perspective our results.

- To know if NP300 and NP650 are representative of the sample, (as the authors suggested), the authors need to quantify it, for example, by adding a histogram obtained with size measurements of the NP from electron microscopy images (average size, dispersion...)

Answer: As shown in the SEM image and the histogram of the size distribution (now shown in Fig. S1), NP300 and NP650 are rather representative of the sample.

•S13:

Modify Start / End which is too confusing.

Answer: We replaced “Start / End” with “First / Last experiment” in new section S16 (old section S13).

• S23:

- It seems that this figure is not showing the conversion of the sample presented in this paper. This should be clearly described in the figure caption.

- Moreover, S23 does not appear in the main manuscript (and is not referred in the Supplemental Materials). As already mentioned, a figure presented should be discussed, otherwise it should not be presented.

Answer: It is now clearly mentioned in the figure caption (now Fig. S24) that the mass spectrometry is from a different sample with Pt particles very similar in size to the sample measured in the manuscript. The Figure is now referred in the main manuscript.

• Fig, 2 (and equivalent in SM):

- Add first/last measurement on the image itself. It would help the reader to understand (faster) what is shown

Answer: We have added first/last measurement in Fig. 2 and in SM.

Other minor corrections in the pdf file.

Answer: We have included all the minor corrections described in the pdf file.

Reviewer #3 (Remarks to the Author):

The authors' response and revision of manuscript addressed my main concern about the measurement nature, which samples strain along a particular direction, *e.g.* (111), in 3D. In authors' reply, the limitation of using the authors' technique for studying smaller nanocrystals was discussed. While this limits the significance about size-dependence, the report represents the state of art and could motivate further experiment on this topic. Publication is recommended.

REVIEWER COMMENTS

Reviewer #1 (Remarks to the Author):

After thoroughly reading the revised version, many concerns have been resolved. I have a few comments.

1. In their label as 'first' and 'last measurement', since they use the different times for each cycle, they should mention how they decide the 'last' measurement for the number of scans for each cycle, e.g., until reaching the saturation of the changes?
2. NPD1 shown in Fig S2 is also 650 nm size. It seems to show smaller values of displacement in the O2 condition. The observation should be compared with the NP650.

Reviewer #2 (Remarks to the Author):

Even though the authors tried to answer all my questions, there are still confusing points:

- 1) The values of the strain field energy presented fig. 3 are different from the ones displayed on the previous version.
~ 0.1 fJ in the new version and 0.8 E-11 J in the previous one.

WHY ????

- 2) In their answer, the authors wrote: "1) These large NPs do present catalytic activity at high temperature as demonstrated in Fig. S24 (mass spectrometry from a different sample with Pt particles very similar in size to the sample measured in the manuscript)."

The authors assume considering the mass spectrometry signal measured on a different sample (with Pt particles very similar in size to the sample measured in the manuscript). that all nanoparticles present on the substrate are participating to the catalytic reaction

However, this MS signal IS NOT demonstrating that the larger NPs do present catalytic activity (neither the opposite). The MS signal allows to conclude that the sample (in a global point of view) is indeed showing a conversion, but there is no clue about which type of NP are responsible of the catalytic properties (small NP, big, NP, all, only a few of them? ...).

REVIEWER COMMENTS

Reviewer #1 (Remarks to the Author):

1. In their label as 'first' and 'last measurement', since they use the different times for each cycle, they should mention how they decide the 'last' measurement for the number of scans for each cycle, e.g., until reaching the saturation of the changes?

Answer: The 'last' measurement was systematically taken after reaching the saturation of the changes. It corresponds to the steady-state of the particle. This is now mentioned in the revised version of the manuscript (see caption of Figure 2).

2. NPD1 shown in Fig S2 is also 650 nm size. It seems to show smaller values of displacement in the O₂ condition. The observation should be compared with the NP650.

Answer: We have inserted a new Supplementary Figure S2 and Table S3 to discuss this point. In the previous Supplementary Figure S2, the color scale of the phase varied from $-\pi$ to π , a larger scale range than the one (± 0.8 rad.) used in Supplementary Figure S7 showing the displacement/phase of NP650. This is due to the fact that NPD1 contains a dislocation with a large displacement field that extends over the whole particle volume. In the new version of the manuscript, we are now displaying the strain field that shows more clearly the surface strain evolution despite the presence of a dislocation. As illustrated in new Fig. S2 and new Table. S3, NPD1 also experiences significant strain relaxation in Ar + 2%O₂ atmosphere, in good agreement with our observations for the defect free NPs. In addition, the initial strain field in Ar atmosphere is relatively consistent with the one observed in NP300 and NP650. The differences with the defect free NPs most likely arise from the presence of the dislocation but also from the difference in shape (presence of a large truncated (0 0 -1) facet in NPD1 for instance). This is now discussed in Supplementary Section S3.

Reviewer #2 (Remarks to the Author):

1) The values of the strain field energy presented fig. 3 are different from the ones displayed on the previous version.

~ 0.1 fJ in the new version and 0.8 E-11 J in the previous one.

Answer: We apologize for this mistake. The correct strain field energy values are the one displayed in the last version (~ 0.1 fJ) and not in the first one (0.8e-11 J). By recalculating the strain field energy, we observed that we did not integrate over infinitesimal voxel sizes but twice over the full volume of the particle. The values (in fJ) are in line with previous publications, see for instance [A. Ulvestad, A. Singer, H.-M. Cho, J. N. Clark, R. Harder, J. Maser, Y. S. Meng, and O. G. Shpyrko, Single Particle Nanomechanics in Operando Batteries via Lensless Strain Mapping, Nano Letters 14, 5123 (2014)].

2) In their answer, the authors wrote: “1) These large NPs do present catalytic activity at high temperature as demonstrated in Fig. S24 (mass spectrometry from a different sample with Pt particles very similar in size to the sample measured in the manuscript).”

The authors assume considering the mass spectrometry signal measured on a different sample (with Pt particles very similar in size to the sample measured in the manuscript). that all nanoparticles present on the substrate are participating to the catalytic reaction
However, this MS signal IS NOT demonstrating that the larger NPs do present catalytic activity (neither the opposite). The MS signal allows to conclude that the sample (in a global point of view) is indeed showing a conversion, but there is no clue about which type of NP are responsible of the catalytic properties (small NP, big, NP, all, only a few of them? ...).

Answer: We agree with the referee that the MS signal is not demonstrating that the larger NPs do present catalytic activity (neither the opposite). It is the structural (strain) changes while changing the gas mixture, even if there are weaker for NP650, which indirectly demonstrates that these particles participate in the reaction as they evolve with gas.

REVIEWERS' COMMENTS

Reviewer #2 (Remarks to the Author):

The authors' response addressed my main concerns.